# Light and Plant Growth Regulators on In Vitro Proliferation

**DOI:** 10.3390/plants11070844

**Published:** 2022-03-22

**Authors:** Valeria Cavallaro, Alessandra Pellegrino, Rosario Muleo, Ivano Forgione

**Affiliations:** 1Institute of BioEconomy (IBE), National Research Council of Italy, 95126 Catania, Italy; alessandra.pellegrino@cnr.it; 2Tree Physiology and Fruit Crop Biotechnology Laboratory, Department of Agriculture and Forest Sciences (DAFNE), University of Tuscia, 01100 Viterbo, Italy; ivano.forgione@unitus.it

**Keywords:** light spectra, light fluence rate, photoperiod, growth regulators, in vitro culture

## Abstract

Plant tissue cultures depend entirely upon artificial light sources for illumination. The illumination should provide light in the appropriate regions of the electromagnetic spectrum for photomorphogenic responses and photosynthetic metabolism. Controlling light quality, irradiances and photoperiod enables the production of plants with desired characteristics. Moreover, significant money savings may be achieved using both more appropriate and less consuming energy lamps. In this review, the attention will be focused on the effects of light characteristics and plant growth regulators on shoot proliferation, the main process in in vitro propagation. The effects of the light spectrum on the balance of endogenous growth regulators will also be presented. For each light spectrum, the effects on proliferation but also on plantlet quality, i.e., shoot length, fresh and dry weight and photosynthesis, have been also analyzed. Even if a huge amount of literature is available on the effects of light on in vitro proliferation, the results are often conflicting. In fact, a lot of exogenous and endogenous factors, but also the lack of a common protocol, make it difficult to choose the most effective light spectrum for each of the large number of species. However, some general issues derived from the analysis of the literature are discussed.

## 1. Introduction

Plants, like any other living organisms on planet Earth, are strongly influenced by environmental cues. Unlike animals, plants are sessile and at the mercy of their surrounding environment. Consequently, they have evolved mechanisms that perceive and respond to environmental changes and adapt their development and growth accordingly. Light plays a pivotal role in a plant’s life, not only for photosynthetic energy production but also for its regulative role of molecular, biochemical and morphological processes that underlie plant growth and development [1,2,3]. Fluence rate, regions of wavelength electromagnetic spectrum, duration and direction are the key attributes of light that drive photosynthesis and photomorphogenesis through mechanisms including the selective activation of various light receptors [4,5,6,7,8,9]. Plant light photoreceptors have evolved in articulated biochemistry structure that capture photons and detect many of the light physical properties. Subsequently, through interactive pathways the photoreceptors interpret information from incoming light and traduce them in biochemical and biological responses able to regulate plant growth and development. A discrete number of photosensor families have evolved in plants. The phytochrome (PHY) family receptors monitor the red (R, 600–700 nm) and far red (FR, 700–750 nm) light regions [10,11,12]. PHY can be present in two states and the active state (P_fr_) is formed due to absorption of red light by the inactive state (P_r_) [13]. The wavelength region of light from UV-A to blue (B, 320–500 nm) is perceived by three small families of photoreceptors [14] that mediate plant responses. All three photoreceptor families contain flavin adenine dinucleotide (FAD) as a chromophore: three cryptochromes (CRY) with CRY1 and CRY2 acting in the nucleus, whereas CRY3 is probably acting in the mitochondrion and chloroplast [15,16], two phototropins (PHOT) [9,11,17] and the members of the *Zeitlupe* family (ztl, fkf1 and lkp2) [18]. In addition, PHY has also been found to mediate various blue responses [19]. The UV Resistance Locus 8, monitoring ultraviolet B wavelengths (UV-B, 280–315 nm), regulates both developmental and UV-protective outcomes [20,21,22].

PHYs act in detecting mutual plant shading through the change in the R:FR ratio and appropriately redirect growth and development through the modulation of apical dominance and of axillary meristems formation according to survival [23,24,25]. CRY1 is thought to be the CRY responsible for the B high-irradiance response, inhibiting stem plant growth and reducing internode elongation, whereas CRY2 is likely responsible for the inhibition because of the B low-irradiance response [19]; collectively, in plants they perform important traits such as flowering and plant stem elongation [26]. PHOT1 and PHOT2 are involved in auxin polar transport, modulation of auxin sensing and phototropism [27,28,29].

*Micropropagation* is considered an effective large-scale in vitro plant multiplication of selected insect/disease/virus-free plants in a short time, all year round, and is a reliable method for in vitro preservation of threatened plant species. The micropropagation technology differs strongly from all other agamic propagation methods since the plants, cultured frequently as microcuttings, can remain under constant environmental conditions for a long time. The habitat of an in vitro culture is strongly restricted, and plants switch from an ontogenetic processing that starts from similar juvenility traits to a much deeper juvenility state [30]. Photoperiod, light intensity, light quality, temperature and relative humidity are factors that in the in vitro habitat are subjected to scarce fluctuations that alter the periodic and oscillator systems upon which plants depend; therefore, plants remain under largely invariable conditions. Although, currently, we cannot establish whether the mutations that are detected in the genomes of in vitro growing plants appear during in vitro culture, however, we could hypothesize that under pressure of these unnatural conditions, plants develop adaptive mechanisms to survive in limited spaces. These adaptive mechanisms involve epigenetic modifications that are programmed to confer plasticity to in vitro plants [31].

Tissue culture is also used in genetic improvement procedures with the aim of selecting subjects under the conditions of selected stress pressure, although in most cases the conditions do not reproduce the real ones. Evolution, in fact, diversifies and adapts species to better achieve suitability to the environmental conditions prevailing at a given time and habitat; a chain of genetic adjustments is selected at the same time as the periodic physiological events that generally occur during plant’s life [32].

In vitro propagation proved to be particularly valuable for vegetatively propagated plants such as *Solanum tuberosum* L., *Allium sativum* L., *Musa acuminata*, *Saccharum officinarum* L., different ornamentals, orchids and fruit trees and energy crops [33,34]. Currently, micropropagation has also attracted growing attention from researchers as an efficient alternative way for rapid and controlled production of bioactive phyto-chemicals or food ingredients from medicinal and aromatic plants.

However, the effectiveness of a micropropagation protocol depends on the proliferation rate and stability, i.e., the number of explants, such as microshoots and single nodes, obtained from a single donor plant [35]. In addition, adventitious roots induction and the subsequent extra vitro acclimation of plantlets determine the success of a commercial propagation protocol [2]. The multiplication of shoots is based on the concomitance of two iterative processes: the induction and formation of phytomer, which includes lateral meristems formation (axillary buds) from the apical meristem (apex) and the subsequent outgrowth of the axillary buds into new shoots [36]. In this contest, artificial light plays a crucial role in successful in vitro plant production, together with other factors such as medium composition, gas exchange in the culture vessel, temperature and specific physiological outcomes of plant explant, i.e., the species-specific physiologic adaptation to the in vitro conditions previously described. Illumination should provide light in the appropriate spectral regions for promoting photosynthetic metabolism and photomorphogenic responses [37,38]. Controlling light quality (wavelength ranges), irradiances (photon flux) and light regime (photoperiod) enables the production of plants with desired characteristics [35,39].

From the outset, the lighting systems used in in vitro plant growth had been fluorescent tubes (Fls), high pressure sodium (HPS), metal halide (MH) and incandescent lamps (IL) with varying wavelengths from 400 to 700 nm. Among these, Fls have been the most popular in tissue culture rooms and consume approximately 65% of total electricity in tissue culture labs [40]. The Fls have high amounts of photons in the infrared and red ranges, gradually dropping toward blue. Due to the presence of phosphor coating, white FLs also have a continuous visible spectrum with peaks near 400–450 nm (violet-blue), 540–560 nm (green-yellow) and 620–630 nm (orange-red). The main inconveniences tied to the use of these lamps are: (i) a significant portion of the spectral output emitted (from 350 to 750 nm) [41] is not utilized by the plant cultures since they are abundant in green (G) and yellow (Y) light, which are less efficient for plants and usually lack FR light [35,41], (ii) light irradiation may cause photo-inhibition of growth and differentiation [42] and photooxidative damage in plants [43] and (iii) the dissipation of a large amount of energy as heat [44].

In recent years, light-emitting diodes (LEDs) have attracted increasing attention as potential light sources for various applications of plant tissue culture [40]. The advantages of LED lights over conventional lighting systems mainly consist in the higher photosynthetic photon efficacy (PPE) as compared to the previously used HPS or Fls. The maximum PAR efficiency of LED lamps ranges between 80 and 100%, while Fls provide only 20–30% [45,46]. The precision in converting electrical energy to photons of specific wavelengths at the desired photosynthetic photon flux density (PPFD) with negligible heat loss makes LEDs more energy-efficient than all other available artificial lighting sources. Based on the manufacturers’ specifications, the LED lamps require about 32% less energy than the Fls per μmol m^2^ s^−1^ of photons delivered to the plants [34] and 10–25% total energy saving can be realized when considering climate modification by the transition from HPS to LED [47]. Moreover, LED lamps possess a longer operating lifetime (>50,000 h), negligible heat emissions and, consequently, an indirect reduction in refrigeration costs, a more robust and easy-to-handle plastic body, no emissions of greenhouse gases (CO_2_) for their production and they produce no mercury pollution [46,48].

The narrow waveband emission and dynamic control of light intensity in LED-based illumination systems allow the choice of spectral quality to match the absorption range of a specific photoreceptor and thus to regulate the photosynthetically and photomorphogenic responses required for the cultivation of each species in vitro [41]. For these reasons, the use of LED lamps in the in vitro culture systems is a useful tool for photobiological studies since they allow the control of irradiance and the emission of specific spectral patterns [41]. With the rapid advancement of the technology, the reduction of LED prices and the diverse studies that show more vigorous in vitro plants cultivated under these lighting conditions, the replacement of Fls with LED lamps has attracted considerable attention around the globe [9].

Numerous studies reported the applications of LEDs, as compared to white Fls, in promoting in vitro organogenesis, growth and morphogenesis from various plant species such as *Gossypium hirsutum*, *Anthurium andreanum*, *Brassica napus*, *Musa acuminata* and so on [49,50,51,52]. The impact of LED lighting on somatic embryogenesis has also been explored for a few plant species [53,54,55,56,57,58].

Although there are a discrete number of studies, many tissue culture laboratories hesitate to replace conventional lighting systems with LEDs out of apprehension of an unpredictable and aberrant in vitro, which may damage consolidated production protocols [59].

Moreover, light quality influences the biological effectiveness of the growth regulators added to the culture substrate, as well as the endogenous hormonal balance in the tissues [60], which must be readdressed after the substitution of the old ones with LED lamps.

Keeping this in mind, in this review, the attention will focus on the literature on the effects of light on shoot proliferation, a main process of in vitro propagation. The effects of the light spectrum on the balance of endogenous growth regulators will be also presented.

## 2. Effects of Spectral Quality of Light on In Vitro Proliferation

The spectral quality of light significantly influences the shoot biological response. Since plant photoreceptors responsible for plant development and photosynthesis are known to be primarily and most significantly stimulated by red (RL) and blue (BL) regions of the light spectrum, most of the studies evaluated the impact of monochromatic RL (660 nm), BL (460 nm) and combined BL (440–480 nm) with RL (630–665 nm) lights. Scarce is the information available on the effects of the far-red (FRL), green (GL) and yellow (YL) regions of the spectrum [44]. For each light spectrum, the evaluated effects concern the proliferation rate and characters related to development, morphology and plantlet quality, i.e., shoot length, fresh and dry weight and photosynthetic pigment accumulation. In fact, the light treatments yielding higher chlorophyll and carotenoid contents (relevant components of the light-harvesting antenna of photosystems) are generally linked with improved fresh and dry matter accumulation and shoot growth [50,61,62,63,64,65,66]. The main results obtained on flowering plant species are presented in Table 1 and Table 2.

### 2.1. Red Light Effects

#### 2.1.1. Red Light Effects on Shoot Proliferation

Some authors agree on the positive role of RL [123], and high-ratio RL:FRL [35] on shoot proliferation [135]. RL significantly enhanced the adventitious bud formation and development in *Gerbera jamesonii* [136], in *Lactuca sativa*. [137], in *Spathiphyllum cannifolium* [83], in *Stevia rebaudiana* [114] and in *Mirtus communis* [120]. RL was effective for bud formation and outgrowth in *Pseudotsuga menziesii* embryo cultures [122]. In contrast, as compared to the cultivation under WL or combined RL with BL, under monochromatic RL or BL, Bello-Bello et al. [106] observed a decrease in the proliferation ratio in *Vanilla planifolia* Andrews and Estrada et coll. [111] and Lotfi et al. [59] found the same decrease in *Anthurium andreanum* and in *Pyrus communis* L., respectively. Somatic embryo germination and conversion of three southern pine species [53] and *Cydonia oblonga* [126] were positively affected by application of RL.

Positive effects of RL illumination have been ascertained in many orchids. In *Cymbidium Waltz* ‘cv Idol’, the highest protocorm-like bodies (PLBs) formation rate (100%) was found in the culture media containing 0.01 and 0.1 mg L^−1^ N- acetylglucosamine (NAG) under RL, although a promotive role was observed under GL, but at 1 mg L^−1^ NAG [100]. In a study of Mengxi et al. [90], the highest PLBs induction rate, propagation coefficient and fresh weight of *Oncidium Gower Ramsey* were observed under RL treatment, which agrees with observations on the Cattleya hybrid [138]. However, in this last species, monochromatic RL resulted in an impaired leaf growth and chlorophyll content. Moreover, in *Oncidium Gower Ramsey*, even if R-LEDs promoted PLB induction, it was observed that BL emitted by LEDs promoted a differentiation of PLBs [90]. Hamada et al. [88] found that R fluorescent lamps increased the PLB proliferation of *Cymbidium finlaysonianum*, even if used only during the early stage of the culture. The R spectrum was effective for *Cymbidium* callus proliferation [80] but not for the successive propagation. The combination of RL and FRL wavelengths determined the highest number of somatic embryos in *Doritaenopsis ‘Happy Valentine’* [54].

The action mechanisms promoted by RL has been investigated by different authors. In *Vitis vinifera*, the axillary shoot development could be due to the release of apical dominance caused by BL, as suggested by Chée [68] and Chée and Pool [70]. Similarly, Burritt and Leung [79] observed that the inhibitory influence of FRL on shoot proliferation is reversible, whereas exposure to BL permanently reduces explant’s competence for new shoot formation. They suggested that PHY and an independent BL photoreceptor, probably CRY, regulate shoot production from *Begonia × erythrophylla* petiole explants. RL has been shown to exert effects on plants proliferation through the PHY, which, in the active form, would alter the endogenous hormonal balance increasing in the quantity of cytokinin (CK) in tissue, counteracting the action of auxin and thus determining an increase in the development of lateral shoots [139,140].

Moreover, research on the effects of PHY on in vitro multiplication of shoots of the *Prunus domestica* rootstock GF655-2 [141] demonstrated that the actions of WL, BL and FRL on shoot proliferation were fluence-rate dependent, while RL was effective both at 37 μmol m^−2^ s^−1^ and at 9 μmol m^−2^ s^−1^. The increase in light intensity had, instead, a positive effect on the production of axillary shoots in a *Prunus domestica* Mr.S.2/5 shoot exposed to RL and BL. However, if the number of shoots produced was expressed as a percentage of the total number of axillary buds, the rate of bud outgrowth for each shoot under RL was significantly higher than that detected under BL [142].

The effects of RL on proliferation are also largely dependent on the growth regulators, mainly cytokinins (CKs) applied to the culture medium, and they were found to be indispensable in the outgrowth of lateral buds in *Prunus domestica* rootstock shoots [142]. The same was true for *Spiraea nipponica* where the interaction between CKs and RL resulted in an enhancement of the shoot proliferation rate [123]. Plantlets of this species exposed to RL and FRL resulted in more marked growth than under WL [123]. Interesting interactions resulted from the growth of this species under low RL:FRL photon fluence followed by high-fluence WL and the benzyl aminopurine (BA) levels [123]. More detailed information on the interactions between light and growth regulators will be provided in paragraph 5.

#### 2.1.2. Red Light Effects on Shoot Morphology

Stem elongation, leaf growth and chlorophyll reduction are frequently observed under RL and are all supposed to be associated with shade-avoidance syndrome (SAS) [8].

*Shoot and internode elongation:* It is mostly reported that RL enhances the elongation of primary and axillary shoots when there is an actively growing apex [74,75], and it determines changes in the plant anatomies [143] of multiple species [36]. The RL effect on stem elongation is species dependent. RL increases shoots and internode lengths in *Pelargonium × hortorum* [144], *Vitis vinifera* [85,145], *Rehmannia glutinosa* [65,146], *Gerbera jamesonii* [118], *Abeliophyllum distichum* [98], *Vaccinium ashei reade* [110,147], *Ficus benjamina* [94], *Cymbidium spp*. [148], *Plectranthus amboinicus* [48] and *Fragaria × ananassa* plantlets [149]. The promotive effect of RL was also found on the elongation of secondary and tertiary shoots of *Malus domestica* rootstock MM106 [128], and on in vitro zygotic embryo germination and seedling growth in chestnut, whereas BL suppresses them [150]. In *Populus americana*, cultivar ‘I-476′, shoot length and leaf area of in vitro plants were greatest when exposed to RL, whereas on the other poplar cultivar, ‘Dorskamp’, BL plus RL were more effective [131]. An increase in the shoot elongation caused by internode elongation under red LEDs may result in stem fragility because of excessive elongation of the internode, as occurred in the third internode from the apex of *Dendranthema grandiflorum* Kitam cv.Cheonsu [42] and in *Rehmannia glutinosa* [146]. Following these results, it is required to adjust the ratio of RL when mixed with BL or Fl. In *Fragaria × ananassa* under R-LEDs, leaf petioles were elongated but the leaves turned yellowish green, revealing an irregular in vitro growth [149].

RL also caused thin elongated shoots and the formation of small leaves in *Solanum tuberosum* cv. Miranda, while BL produced short shoots with regular leaf development and many micro-tubers. The micro-tuber development was reversed when the IAA was added to the medium [71]. According to Kim et al. [42], synergistic interactions among CRYs and PHYs may promote or inhibit stem elongation in various ways in different species.

Differences in the response of the different species in the response to the RL:FRL ratios may be explained by the different habitats in which the species evolved. It has been proposed from studies on the elongation of shoots of *Vitis vinifera* [70], *Disanthus cercidifolius* and *Crataegus oxyacantha* axillary shoots [75] that this enhancement is PHY-mediated through the control of enzyme-affected auxin degradation, such that the extremely photolabile auxin would be conserved in cultures illuminated with RL and degraded in cultures under BL. In addition, other plant hormones may be modulated by light and by PHY directly (see paragraph 5).

*Fresh and dry weight:* The greatest mean fresh and dry weight of each cluster of the *Malus domestica* rootstock M9 was observed under RL and it was 83% greater than that observed under WL [135]. Gains in fresh weight were observed in *Vaccinium ashei* [110] and *cattleya* [138]. Dry weight was positively affected by RL in *Myrtus communis* L. [120], in *Euphorbia milii* and *Spathiphyllum cannifolium* [83] and in *Plectranthus amboinicus* [48]. Furthermore, increased growth of in vitro cultured plants provided by RL was also shown in *Scrophularia takesimensis* [102], *Lippia gracilis* [119] and *Vitis vinifera* [145]. Likewise, dry weight increased under RL, probably by the promotion of starch accumulation [50].

*Chlorophyll content:* R-LED increases chlorophyll content in *Musa acuminata* [52], *Passiflora edulis* [151] and *Rehmannia glutinosa*, although less than B-LED [65]. Most authors agree that RL, as compared to other light spectra, promoted leaf growth [74,131,152] but decreased the chlorophyll and carotenoids content of in vitro plantlets [83,90,148,153,154]. On the contrary, Cybularz-Urban et al. [138] found that in *Cattleya* plantlets grown in vitro RL caused the collapse of some of the mesophyll cells and a reduction of leaf blades, meaning that, in the absence of BL and/or WL/GL, the regular development of cells and leaf tissues is blocked. Similar results were found in cultures of birch [154] where the total content of chlorophyll under BL was twice that detected under RL. Smaller amounts of chlorophyll a and carotenoids were also detected in cultures of *Azorina vidalii* [74] under RL, FRL and RL:FRL. Other authors wrote that prolonged RL illumination may result in the ‘RL syndrome’, which is characterized by low photosynthetic capacity, low maximum quantum yield of chlorophyll fluorescence (Fv/Fm), carbohydrate accumulation and impaired growth. It was observed, also, that thylakoid disarrangement in the chloroplast is proportional to the increasing incidence of RL [155]. This damage may be reduced by adding BL to the light spectrum [156]. Regulation of carbohydrate metabolism by light quality has been well documented [41,157]. RL emitted by LED seemed to promote the accumulation of soluble sugar, starch and carbohydrate in upland *Gossypium hirsutum* L. and *Brassica napus* [50,51,158] and in *Oncidium* [16,87]. RL probably may inhibit the translocation of photosynthetic products, thereby increasing the accumulation of starch [50,154]. Moreover, Li et al. [50] suggested that plantlets with lower chlorophyll content utilize the chlorophyll more efficiently than plantlets with higher chlorophyll content under R-LEDs.

### 2.2. Blue Light Effects

#### 2.2.1. Blue Light Effects on Shoot Proliferation

The effects of BL are often reported to be antagonistic of RL ones, although the studies reported in literature concerning the role played by BL on new meristem formation are not always consistent. The positive effects of BL on the stimulation of shoot production and growth of *Nicotiana tabacum* during in vitro culture were reported, but at a higher light intensity [67], and the authors hypothesized photoinactivation of IAA. Five weeks of exposure to BL induced the highest shoot production from *Nicotiana tabacum* callus [159]. Monochromatic BL increased shoot number in *Ficus benjamina* [94], the number of shoots and nodes in *Vitis vinifera* L. *hybrid* [68,70], the number of adventitious buds in *Hyacinthus orientalis* L. [160] and the percentage of organogenesis and the mean number of buds per explant in *Curculigo orchioides* [103]. Higher percentages of BL in the light spectrum were also effective on in vitro shoot induction and proliferation of *Anthurium andreanum* [49], *Gerbera jamesonii* ‘Rosalin’ [107], *Remnania glutinosa* [65] and *Saintpaulia ionantha* [69]. In various species, positive results on proliferation from adding different ratios of B to the R spectrum have been described and will be widely discussed in sub-paragraph 2.3.1. The proliferation rate was greater in *Brassica napus* plantlets when cultured under monocromatic BL and BL plus RL [51]. In lavandin, on a BA-free medium, shoot number was enhanced under BL, WL and RL at low photon fluence rates [72]. In *Oryza sativa* [121] under B-LED illumination, the time required for callus proliferation, differentiation and regeneration was the shortest and the frequency of plantlet initiation, differentiation and regeneration was the highest. Concerning orchids, in *Dendrobium officinale*, the monochromatic BL and RL:BL (1:2) emitted by LEDs determined a higher percentage of protocorm-like bodies (PLBs) producing a higher number (1.5 fold) of shoots [92], in *Cattleya intermedia × C. aurantiaca* the number of shoots regenerated from PLBs was enhanced by BL [161]. In *Oncidium*, RL promoted PLB induction from shoot apex and the higher content of carbohydrate but the lowest differentiation rate, while the highest differentiation rate and protein content were observed under B-LED [87]. BL increased node and total shoot number as compared to RL, FRL and dark in *Prunus avium* cv ‘Hedelfinger’ and one of its somaclones [127]. In contrast, on *Begonia erythrophylla* petiole explants, RL played a role in meristem initiation and BL and FRL were antagonistic to meristem formation, but BL was important for primordia development [79]. In *Gerbera jamesonii* [118], inhibition of shoot multiplication and a reduced plant height was observed under BL compared to what resulted from all other light treatments, and a decrease of lateral shoots number was observed on *Malus domestica* [135] as compared to RL. The same study demonstrated that BL inhibited the rate of proliferation, increasing the apical dominance. Inhibition of meristematic tissue proliferation by BL has also been observed for the embryogenic tissue of Norway spruce [162]. The conflicting reports found in the literature might not only be attributed to species effects, but also to the different types of explants and to the stage of the organogenic process. Hunter and Burritt [81], working on different *Lactuca sativa* L. genotypes, observed a significant decrease under monochromatic BL in shoot proliferation as compared to RL or WL. They argued that RL is required for the formation of shoot primordia, whereas BL is inhibitory to primordia initiation. The effects of RL and BL on this species depended on the stage of the organogenic process in which *Lactuca sativa* plantlets were exposed to the different lights. Exposure to BL during the critical first few days of culture, when meristems are being initiated, results in a significant reduction in the number of shoots produced as compared to exposure to RL and WL. Furthermore, this suppression of meristem initiation is permanent and not reversible afterward by culturing plants under RL. Observations with a scanning electron microscope (SEM) clarified that the lowest shoot development under BL was attributable to the production of much more callus as compared to those cultured under WL or RL, demonstrating that rapid cell division occurred, although the organized center of cell division required for primordia formation was reduced. Moreover, the same authors observed that explants exposed to continuous RL developed numerous small shoot primordia, which occurred more slowly than those detected on tissue exposed to WL. Based on the literature, they stated that the stimulatory effects of RL as compared to WL is genotype dependent, but the inhibitory effect of BL is more widely diffused. Callus formation as affected by continuous BL illumination was observed also in *Pyrus communis*, where callus weight doubled as compared to BL plus RL and BL plus FRL [59]. In *Ficus benjamina*, BL induced a huge formation of callus at the basal section of shoots [94]. Other studies have shown that the timing of exposure to different light regimes is also critical for shoot development *in vitro*. For example, at least 2 wks under RL were required to improve shoot numbers from *Pseudotsuga menziesii* callus, and the length of time in which RL promoted shoot production lasted only 2–3 wks [122]. It was suggested that PHY plays an inductive role in organogenesis of *Lactuca sativa* L., as suggested by Kadkade and Seibert [137], in contrast to antagonistic role of BL, probably via CRYs.

In a series of research projects carried out with different rootstocks of *Malus domestica*, *Prunus domestica* and *Prunus persica*, M9, MM106, Mr.S.2/5, and GF677, respectively [125,128,142,163], it was demonstrated that BL induced, in the starting explant and in the developed shoots, a greater number of nodes with shorter internodes than those observed in RL and in dark. It should be noted that the percentage of nodes that formed lateral shoots was higher in the presence of RL as compared to the BL one. In the *Malus domestica* M9 rootstock, the percentage of sprouted buds under RL was double that under BL [135].

Based on these results, shoot multiplication can be defined as the result of two events: the induction and formation of new buds from the apical meristem and their sprouting through the reduction or the suppression of apical dominance [2,36]. BL would increase the number of axillary buds but, in contrast, it exerts an inhibitory action on buds sprouting (increase in apical dominance). RL, on the other hand, would reduce the apical dominance even though it reduces the formation of new axillary buds. The lower outgrowth of buds in the presence of BL compared to RL would indicate a role in a specific photoreceptor(s) of BL, which would act as an antagonist of the PHY. Photomorphogenetic events detected in the presence of RL and BL would agree with an antagonistic model of stem branching, modulated by light through the PHYs and the photoreceptors of BL, which would interact with each other according to a dynamic model. Moreover, Muleo et al. [142] also showed that the internode extension inhibition under BL exposure and the concomitant positive effect of BL in enhancing axillary bud formation (neoformed nodes) was dependent on the photon fluence rate, but not on PHY photoequilibrium or on concomitant exposure to RL. A quantitative BL threshold was found near 30 µmol m^−2^ s^−1^ (400–500 nm); up to this value, internode extension decreased [142].

Plants, thus, possess a complex and dynamic light response and memory system that involves reactive oxygen species and hormonal signaling, which are used to optimize light acclimation and immune defenses [164]. Thus, regulating the spectral quality, particularly by the B-LED, improves the antioxidant defense line and is directly correlated with the enhancement of phytochemicals in *Rehmannia glutinosa* [65]. Mengxi et al. [90] found higher values of superoxide dismutase (SOD), peroxidase (POD) and catalase (CAT) activities in leaves under B-spectrum irradiance and concluded that B-LED may be more satisfactory for activating different defensive systems to reduce excessive amounts of reactive oxygen species. However, in two important *Dianthus caryophyllus* cultivars, ‘Green Beauty’ and ‘Purple Beauty’, RL treatment also increased the activities of antioxidant enzymes and nutrient contents [165]. The B-LED illumination also significantly increased the antioxidant enzyme activities in leaves and roots in *Amaranthus tricolor* and *Brassica rapa* L. subsp. *oleifera* [166]. In the in vitro cultured *Pyrus communis* plantlets, it was detected that the gene encoding the pathogenesis-related protein PR10 is regulated daily by the body clock of a plant, while *PR1* was expressed without clear evidence of circadian regulation [167]. In the same studies, a specific function was played by PHYB and CRY1 photoreceptors, considering that in transgenic plants the first photoreceptor enhanced the gene expression of *PR1* 5- to 15-fold, and CRY1 enhanced plant resistance to the *Erwinia amylovora* bacterial infection [167]. *Prunus avium* rootstock plantlets, overexpressing the PHYA gene and grown in vitro, displayed a strong resistance to bacterial canker (*Pseudomonas syringae* pv. morsprunorum), highlighting a role of light quality and quantity in the regulation of plant resistance to bacterial disease [168]. Therefore, light quality through the regulative network of photoreceptors plays a relevant role in the endogenous rhythms of gene expression and pathogen attacks.

#### 2.2.2. Blue Light Effects on Plantlet Morphology

BL is mostly considered to be able to increase leaf growth, photosynthetic pigment synthesis, chloroplast development and stomatal opening, soluble proteins and carbohydrates and dry matter content and to inhibit stem and root elongation, while RL enhances stem growth and carbohydrate accumulation [41,50,87,158]. In *Scrophularia kakudensis*, BL imposed a stressful environment that resulted in the activation of several proteins related to stress tolerance, photosynthesis, gene regulation, post-translational modification and secondary metabolism [169]. The improvement in the leaf characteristics induced by the addition of BL to RL seem to indicate a better quality of micropropagated plantlets, which in turn may also improve acclimation [2,170].

*Plant height:* A few papers report positive effects of BL on shoot length, while most studies agree on its inhibition of plantlet elongation. The blue spectrum was recognized to inhibit stem growth in *Oncidium* [90], in *Pelargonium × hortorum* [144], in *Dendranthema grandiflorum* [42] and in *Zantedeschia jucunda* [171], especially as compared to RL or RL:FRL. In different tree species, *Prunus domestica* Mr.S.2/5 and *Malus domestica* MM106 and M9, inhibition of internode elongation was also detected [128,135,142]. In contrast, BL (470 nm) and RL (660 nm) illumination were found effective for increasing shoot length in *Achillea millefolium* [172] and *Dendrobium Sonia*, where, however, BL significantly reduced multiplication as compared to YL [116].

In some cases, BL is necessary to contrast the excessive effects of RL on shoot length assuring good plantlet development. Nhut et al. [149] observed that *Fragaria x ananassa* plantlet growth was inhibited under BL, whereas an irregular plantlet growth and development was observed in the absence of BL. In the experiment of Jao et al. [171], a shorter stem of plant and a higher chlorophyll content was found in the RL plus BL treatment, highlighting that BL may be involved in the regulation of both plant height and chlorophyll development.

BL induces the production of short shoots with good leaf development and many micro-tubers in *Solanum tuberosum*. Under BL, kinetin not only strongly stimulated tuber formation, but also increased the total fresh weight and root(+stolons)/shoot ratio [71].

*Fresh and dry weight:* In *Dendrobium officinale*, compared to other light treatments (dark, Fl and R-LEDs), B-LEDs, alone or with R-LEDs (1:2), induced higher dry matter accumulations of PLBs and shoots [92]. Increased biomass production in cultures of *A. millefolium* [172] was noted under monochromatic B-LED or R + B-LEDs. Monochromatic BL determined higher fresh and dry weight and leaf number per plantlets in *Euphorbia milii*, *Spathiphyllum cannifolium* [83] and *Rehmannia glutinosa* [146].

It is noteworthy that monochromatic BL had a negative effect on the dry matter production of *Lippia gracilis* [119], *Plectranthus*
*amboinicus* [48], *Gossypium hirsutum* [50] and *Vanilla planifolia* [106], as well as in the sensitive cv Dopey of *Rhododendron* where it also reduced leaf chlorophyll content [75]. In most cases, however, RL was the most effective in all these species.

Many authors, however, agree on the most positive effects obtained on fresh and/or dry weight of plantlets by adding different ratios of BL to RL as compared to only monochromatic BL (see the next chapter) [62,65,90,173,174]. Moreover, Kurilčik et al. [174] demonstrated that the influence on shoot length and weight of the BL component of a mixed light is tied to the photon flux density (PFD) of the FRL component. Once more, these results indicate the species-specific effects of BL on in vitro plantlet growth [51]. Cioć et al. [120] evidenced the relationship of BL and growth regulators. B-LED illumination and a high BA content in the substrates stimulated the growth of a greater number of *Mirtus communis* L. leaves (BL and RL plus BL) and increased the fresh weight as compared to Fls, but did not affect the dry weight, whereas RL with low amount of BA enhanced both proliferation and shoot growth. Moreover, in *Oncidium*, the amounts of soluble protein in the PLBs and leaves were the highest in the BL treatment, which suggests that the B spectrum was advantageous for protein synthesis [87,90].

*Leaf morphology and functionality:* BL is considered an important regulator of leaf expansion; however, differences have been ascertained among the different species. BL induced the largest number of leaves per plant, and the largest leaf thickness and area in *Altenanthera brasiliana* [175] and *Platycodon grandiflorum* [158] and a similar response on leaf area was demonstrated in *Gossypium hirsutum* [50] and *Brassica napus* [51]. BL enhanced leaf chloroplast area and the translocation of carbohydrates from chloroplasts in *Betula pendula* [154]. In contrast, less leaf area was observed in *Pyrus communis* under monochromatic BL, as compared to RL, RL plus FRL and RL plus BL [59] and in *Azorina vidalii* [74], as compared to RL plus FRL. Furthermore, CRYs are known to regulate chloroplast development in response to BL [176].

*Photosynthetic pigments accumulation:* Several studies have reported that B irradiation resulted in higher chlorophyll contents and carotenoids in the in vitro plantlets as compared to RL and FL. Cultures of *Euphorbia milii* [61], *Doritaenopsis* [63], *Oncidium* [16,87], *Stevia rebaudiana* [114], *Dendrobium officinale* [92], *Prunus avium* cv ‘Hedelfinger’ and in its somatoclone [127], *Zantedeschia jucunda* [171], *Tripterospermum japonicum* [62], *Chrysanthemum* [174], *Anthurium andreanum* [111], *Phalaenopsisis* [177], *Brassica napus* [51] and *Vaccinium ashei reade* [147] exhibited higher total chlorophyll content under monochromatic B-LEDs or combinations of R- plus B-LEDs as compared to cultures exposed to R-LED or Fls treatments. The chlorophyll content, leaf and stomata number per explant were also highest on plants cultured under BL in *Vitis vinifera* [85] and in *Gossypium hirsutum* [50].

BL and UV irradiation enhanced chlorophyll content in *Hyacinthus orientalis* L. [160] and chlorophyll a+b content, but not the carotenoid content, in leaves of *Pyrus communis* [59]. Photosynthetic capacity was highest in *Betula pendula* Roth [154] and in chrysanthemum (*Dendranthema grandiflorum*) [42] when the plantlets were exposed to BL as compared to RL. In *Dendrobium kingianum*, the average number of PLBs and the chlorophyll content were highest under B-LEDs, in contrast to the explants cultured under R-LEDs where the highest shoot formation and fresh weight were observed [99]. Likewise, a study of *Oncidium* PLBs by Mengxi et al. [90] showed that chlorophyll a and b and carotenoid levels and the greatest growth were detected under B-LEDs. On the contrary, a reduction in chlorophyll levels in plants grown under BL was observed in *Vanilla planifolia* [106]. Thus, according to Li et al. [51], the chlorophyll content of in vitro plantlets grown under different light qualities varies within plant species or cultivars. Moreover, even if BL, as compared to RL or different RL:BL ratios, reduced leaf expansion and hence leaf area in *Azorina vitalii*, the chlorophyll and carotenoid content per unit leaf area was higher than RL:FRL [74].

Changes in chlorophyll biosynthesis induced by changes in spectral quality may provide advantages regarding plant growth [178]. The species-specific responses to the B spectrum, in terms of photosynthetic pigments, are probably tied to the different environments in which the different species developed and to the type of explant used for in vitro initiation. In *Lippia gracilis*, plantlets that originated from apical explants had higher pigment production under the BL spectrum, whereas those from nodal explants showed higher production under WL, followed by the BL conditions [119]. These studies indicate that BL provides important environmental information and mostly promotes higher photosynthetic efficiency.

### 2.3. Combined Blue and Red Light Effects

#### 2.3.1. Blue and Red Light Effects on Shoot Proliferation

Many studies have been carried out on the effects of combining BL and RL. A mixture of photon quantity of BL plus RL may combine the advantages of monochromic RL and BL and may overcome the individual disadvantages of these lights. However, a large amount of research regarded the assessment of the best proportion of photon quantity of BL and RL, since different behaviors have been ascertained between species and varieties [50]. In some cases, the same ratio between RL and BL is effective (RL:BL = 1:1); in other cases, higher percentages of RL as compared to BL or vice versa are effective.

A large number of studies demonstrated the promoting role of R- plus B-LEDs in various combinations on shoot regeneration and the growth of the regenerated plants: BL:RL = 1:1 in *Lilium oriental* [78], RL:BL = 9:1 in the recovery of *Solanum tuberosum* plantlets after cryoconservation [97], RL:BL = 9:1 [104] and RL:BL = 7:3 in *Fragaria x ananassa* [149], RL:BL = 7:3 in *Saccharum officinarum* [101] and RL:BL = 1:1 in upland *Gossypium hirsutum* L. [50] and *Abeliophyllum distichum* [98]. In *Gerbera jamesonii* [118], the highest shoot multiplication rate (40% higher proliferation as compared to plantlets grown under Fls) was observed under RL:BL = 50:50 and RL:BL = 70:30. In *Anthurium*
*andreanum*, shoot propagation was promoted by exposure to RL:BL illumination and higher growth under BL [111]. In the same species, following Budiarto [49], the number of regenerated shoots was greater when exposed to higher percentages of B than R-LEDs (RL:BL = 25:75). In *Brassica napus* L. as well, proliferation was greater under higher percentages of BL (BL:RL = 3:1 light, [51]. Good results on shoot proliferation have been also reported in *Azorina vidalii* using high RL and BL combinations (2,3; BL:RL, [74] or high RL:FRL ratios (1,1)). For *Panax vietnamensis* [105], the most effective plant formation was obtained when embryogenic calli were cultured under the combination of 60% RL and 40% BL and was reported to be two times higher than under Fl [105]. Concerning woody species, better results on proliferation were obtained on *Phoenix dactylifera* with an RL:BL ratio equal to 18:2 [133], on *Pyrus communis* with an RL:BL ratio equal to 1:1 [59] and on *Populus x euramericana* with an RL:BL combination of both 70:30 and 50:50 [131] as compared to monochromic lights and Fl.

Concerning orchids, it seems that higher RL percentages as compared to BL ones are effective. A combination of R:B = 9:1 gave the highest shoot proliferation in *Phalaenopsis* protocorms [86]. In *Cymbidium*, 100% R-LED was the most effective for callus induction, but callus proliferation was best under 75% R-LED plus 25% B-LED treatment. PLB formation from callus was obtained in 25% R-LED plus 75% B-LED [80].

The composite light of R- and FR-abundant G2 LEDs (8% BL, 2% GL, 65% RL and 25% FRL-Valoya Oy, Helsinki, Finland) resulted effective in *C. grandiflorum*, *G. jamesonii*, *H. hybrida* and *Lamprocapnos spectabilis* giving similar or higher propagation of the Fls. However, in this case, the influence of FRL and GL must be considered and will be discussed in the following chapters [35].

#### 2.3.2. Blue and Red Light Effects on Plantlet Morphology

Many studies confirmed the effectiveness of R- and B-LEDs in enhancing growth and photosynthesis in many plant species. B- and R-LEDs were developed to grow in vitro plants because chlorophyll a and b show a maximum absorption at their respective wavelengths (460 and 660 nm). The same light ratios were effective on proliferation and in promoting the quality of plantlet characteristics.

*Plantlet elongation:* Various combinations of R- and B-LEDs proved to determine the best results for stem length and leaf growth for *Saccharum officinarum* [112], *Stevia rebaudiana* [114], *Populus x euramericana* cv ‘Dorskamp’ [131], *Pyrus communis* [59], *Fragaria x ananassa* [104] and *Dendrobium officinale* [92]. Sivakumar et al. [179] showed that continuous RL plus BL or intermittent BL significantly stimulated shoot elongation of sweet Solanum tuberosum plantlets in vitro. Hahn et al. [146], on *Rehmannia glutinosa*, found that shoot lengths under either B- or R-LEDs were greater than under mixed LED or Fls, but the plantlets overgrew and appeared fragile, whereas plantlets under mixed LED or Fls were healthy, with normal shoot lengths. Thus, normal plant growth was clearly related to the presence of monochromatic BL or RL. According to some authors, the synergistic interactions between CRY and PHY could either promote or inhibit the shoot elongation in different plant species.

*Plantlet growth:* The composite spectra of R- and B-LEDs positively regulated fresh and, in most cases, also dry matter accumulation. As compared to the cultures raised under Fls or monochromatic lights, in most cases LEDs supplying higher RL ratios (from 70–90%) as compared to the BL ones were effective in enhancing the in vitro growth of different species such as banana [180], grape [145], *Fragaria x ananassa* a [149], *Vaccinium corymbosum* [147], *Tripterospermum japonicum* [62], *Eucalyptus citriodora* [181], *Phoenix dactylifera* [133] and *Lippia alba* [66]. Highest growth was observed under Fl and under a mixture of BL and RL in *Withania somnifera* plantlets [182]. Highest fresh and dry weights were obtained when plantlets were cultured under an equal BL and RL combination (50:50) in different species such as *Chrysanthemum* [42], *Lilium* [78], *Doritaenopsis* [63], *Pyrus communis* [59], *Saccharum officinarum* ([112], upland *Gossypium hirsutum* L. [50], *Vanilla planifolia* [106] and *Solanum tuberosum* [183]. As for proliferation, higher BL rates as compared to the other species are necessary to obtain the best growth in *Brassica napus* [51]. Similarly, to proliferation, higher RL ratios enhanced plant growth and the development of different orchids: *Cymbidium* [148] and *Phalaenopsis* [86]. RL plus BL and FRL or RL plus FRL light significantly enhanced the fresh and dry weights of *Oncidium* plantlets [89].

Differently from other cultures in which the same lights resulted in optimal proliferation and plantlet growth, according to Mengxi et al. [90], in *Oncidium*, the highest induction rate, propagation and fresh weight appeared in the RL treatment, whereas the largest dry weight per plantlet were obtained under B:R = 20%:80% and B:R = 30%:70%, respectively. Differently from other orchids, the in vitro growth of plantlets of the *Calanthe* hybrid was efficiently enhanced under a mixture of BL plus RL (0.7:1) and inhibited by RL plus FRL [184].

*Leaf number and area:* In *Gerbera jamesonii* [118], monochromatic RL and BL treatments resulted in a reduced leaf area, whereas leaf number was enhanced by exposure to RL:BL = 1:1.

R and B mixed LED treatments in various combinations improved leaf number and sometimes length of in vitro cultures of *Fragaria x ananassa* [149] and *Doritaenopsis* [63], leaf area of *Populus x euramericana* [131] and leaf growth of *Stevia rebaudiana* [114].

*Photosynthetic pigment levels*: Many studies showed that optimizing the RL:BL ratio may improve photosynthesis. The positive effect of the appropriate B-:R-LEDs combination on the synthesis of photosynthetic pigments was reported in several studies [51,92]. An appropriate mixture of B- and R-LEDs, compared with solely monochromatic BL or RL, is more effective to increase the chlorophyll a/b ratio and/or carotenoids content of the in vitro grown plants of *Tripterospermum. japonicum* [62], *Lippia alba* [66] and *Staphylea pinnata* [113]. On *Fragaria x ananassa* mixotrophic cultures, the chlorophyll content was the greatest under RL:BL = 70:30 and the least under 100% RL [149].

Plant growth and development caused by increasing the net photosynthetic rate was also observed in *Chrysanthemum* (*Dendranthema grandiflorum*) under mixed R-:B-LED treatments and has been attributed to the adjustment of the spectral energy distribution of RL:BL to chlorophyll absorption [42]. RL or BL plus RL treatments were found more effective in grape for net photosynthetic rates [145] as compared to BL alone. Differences in chlorophyll content in *Artemisia* and *Nicotiana tabacum* plants were ascertained. In plants grown under WL, significantly less chlorophyll content than plants growing in RL:BL (3:1) or RL:BL (1:1) was determined [34]. In *Gossypium hirsutum* L., chlorophyll content, leaf thickness and leaf and stomata area were higher in plantlets cultured under BL; however, the best growth was provided by BL:RL = 1:1 [50]. In addition, in the Colt rootstock of *Prunus avium* exposed to BL and BL plus RL dichromatic light, the leaves had a greater accumulation of chlorophyll [170].

A ratio of BL:RL = 1:1 emitted by LED light facilitated the growth and produced the highest chlorophyll, carotenoid contents and photosynthetic rates in *Oryza sativa* seedlings, but not callus proliferation, differentiation and regeneration, which were enhanced by BL [121].

Different from the other species, higher BL rates as compared to RL (3:1) are necessary in *Brassica napus* L. (cv Westar) to increase chlorophyll concentrations compared to the other LED treatments and Fl. Therefore, the response of chlorophyll content of in vitro plantlets to different light qualities may vary among plant species or cultivars [51].

In different orchid species, BL plus RL was reported as the most efficient treatment on the synthesis of photosynthetic pigments. Shin et al. [63], in *Doritaenopsis*, showed that mixtures of RL plus BL stimulated photosynthesis and chlorophyll accumulation. In *Dendrobium officinale*, chlorophyll a and b and carotenoid contents were the highest in protocorm-like bodies incubated under RL:BL LEDs = 66.6:33.3 [92]. Moreover, in *Oncidium* plantlets, it was demonstrated that the RL and BL combined with FRL or RL plus FRL radiation significantly enhanced chlorophyll content [89].

### 2.4. White Light Effects

#### 2.4.1. White Light Effects on Shoot Proliferation

The use of monochromatic or combined R- or B- LEDs may determine a mismatch with the photosynthetic spectrum. The application of the broad band WL may overcome this problem [44].

*Shoot number:* The best proliferation in *Vanilla planifolia* Andrews [106] was obtained under WL and RL plus BL. Fls and WL increased the *Gerbera jamesonii* ‘Rosalin’ propagation ratio [107]. Similarly, W-LEDs (NS1 lamps of Valoya Oy, Helsinki, Finland) determined by the combination of 20% BL, 39% GL, 35% RL, 5% FRL and G2 LED lamps, enriched in RL and FRL, were as effective as Fls on shoot propagation of *Gerbera jamesonii*, *Heuchera × hybrida*, and *Lamprocapnos spectabilis*. In the same study, the propagation ratio for *Ficus benjamina* was significantly higher under Fls as compared to all tested LEDs. These positive results were attributed to the absence of UV or cool light in the LEDs [35]. Similarly, the most positive effects of Fls on propagation were observed in *Saccharum officinarum* [112] and in *Spathiphyllum cannifolium*, where, however, high citokinins (3 mg L^−1^ BA) were applied [83]. White LED exposure improved the shoot proliferation as compared to Fls but also to RL or RL plus BL lamps in *Musa* spp. [130], *Bacopa monnieri* [109] and *Malus domestica* genotype MM106 [128]. An exposure to low-level WL after 10 days in the dark (to induce organogenesis) determined the regeneration of well-proportioned shoots within 3–4 weeks in transgenic *Petunia x atkinsiana* [77]. In *Prunus domestica* subsp. *insititia*, however, the effect of the light differed in relation to the concentration of CK applied. At the optimal BA concentration (2.7 mM), WL (66 μmol m^−2^ s^−1^) provided better responses on proliferation than RL, BL and FRL, if the CK concentration was below the optimal level, the production of axillary shoots was greater in the RL. The highest BA concentration (13.3 mM) decreased proliferation in monochromatic lights, as BL, RL and FRL, but not in WL [141].

The regeneration of buds from cotyledons of *Lycopersicon esculentum* was high under continuous RL and WL [69]. In *Anthurium* [111], proliferation obtained in WL was similar to Fl. Muleo and Thomas [125] working on *Prunus cerasifera*, obtained better effects on shoot proliferation in intact microcuttings (with apical bud) under WL. Although apical dominance was weakest in the RL and FRL treatments, the highest proliferation of new shoots was detected under WL because of the shorter internodes and high number of new nodes in that treatment as compared to RL, FRL and dark [125].

In contrast, WL, which establishes a similar P_fr_/P_tot_ ratio to RL, did not reduce apical dominance compared with dark. WL would also excite blue-absorbing photoreceptors and the effects of BL on apical dominance were similar to those of WL. It seems, therefore, that the cytokine ratio may be enhanced in woody species under WL to obtain higher proliferation; however, in some species, after a long cultivation time under WL the rate of newly formed sprouts was reduced regardless of the cytokinin concentration but increased when plantlets were exposed to RL [2]. Moreover, under a low BA addition to the substrate (0.5 mg L^−1^), after one month permanence under an R-enriched light (12% BL, 19% GL, 61% RL and 8% FRL), significant enhancement in shoot proliferation in *Ananas comosus* was observed after it was transferred under WL (Cavallaro et al. unpublished data). More than one cycle permanence under the enriched RL, however, determined callus formation on the basis of the shoots, the loss of leaves and impaired growth in *Euphorbia milii* and in *Ceratonia siliqua* L. [185].

#### 2.4.2. White Light Effects on Plantlet Morphology

In *Phalaenopsisis* and *Anthurium andreanum*, treatments with Fls, W-LEDs (460 and 560 nm) and the combination of B- and R-LEDs showed the greatest plantlet length and number of leaves [177]. Shoot fresh and dry weight, plant height, number of leaves, number and length of roots were greater under Fls and W-LEDs in *Vanilla planifolia* [106].

Enhanced chlorophyll biosynthesis was also noted in *Vanilla planifolia* [106] and in different *Saccharum officinarum* varieties [101,112] under W-LED illumination. Exposure to WL was also beneficial for the accumulation of carotenoid pigments in *Saccharum officinarum* [112]. For the apical and nodal segments of *Hyptis suaveolens*, the best growth parameters were provided by W-LED light and RL:BL combinations [186].

### 2.5. Green Light Effects on Shoot Proliferation and Plantlet Morphology

GL has received less attention from the scientific community because it is a misconception that GL mainly plays a role in stomatal regulation, driving photosynthesis through chloroplast gene expression and so contributing to carbon gain. GL’s role in plant growth and development was controversial because it was supposed that, in conveying information, physiological responses were scarce. Since photons of the RL and BL spectrum are depleted by the absorption of plant tissues, the light reflected from and transmitted through the tissues is enriched in photons of the GL wavelength region that efficiently penetrate farther into the body of a plant [187]. Under this condition, GL carries signals for acclimation to irradiance on a whole plant, providing information for fine-tuning developmental acclimation to shade and acting as a secondary antagonistic regulator to the well-known RL:FRL and BL responses [188]. Unlike for RL and BL, a green-light-specific photoreceptor has yet to be discovered [189]. The most accredited GL sensor is the CRY-DASH, which reverts the physiological effect of CRY [190] because many physiological responses regulated by CRY are reversible by GL [191]. Tanada [192] hypnotized the existence of the heliochrome, an FRL:GL reversible receptor acting in complement to PHY. Therefore, GL effects share several attributes that are specific to the receptor antagonists of the physiological actions of RL or BL photoreceptors [128,135,193]. Consequently, GL penetration of the plant canopy potentially increases plant growth by increasing photosynthesis of the leaves in the lower canopy more efficiently than either BL or RL [194].

GL positively influenced shoot branching on the first- and second-order branches of Mr.S.2/5 *Prunus domestica* rootstock and determined a higher internode number and shoot elongation in GF677 *Prunus persica* rootstock [142]. Based on these results, Morini and Muleo [2] hypothesized that GL had a negative effect on apical dominance, similar to RL and YL.

Kim et al. [195] reported that adding 24% of GL to R- plus B-LEDs illumination increased *Lactuca sativa* L. biomass by 47%, even if the total PPFD was the same in both lighting treatments. They attributed the growth-stimulation effect of GL on its ability to penetrate deeper into leaves and canopies. In *Achillea millefolium*, the concentrations of chlorophyll a, chlorophyll b, b/a ratio and carotenoids were higher in plantlets under GL. The highest levels of pigments observed in the GL may indicate plant stress, which can be a way to compensate for the lack of photosynthetically active light [172].

In a study on the *Cymbidium insigne* orchid, the highest PLB formation, shoot formation rate (90%) and root formation rate (50%) were found among explants cultured in a medium supplemented with 0.1 mg L^−1^ chitosan H under GL. After 11 weeks of culture, the fresh weight of PLBs was higher in the treatment with hyaluronic acid (0.1 mg/L) under GL [93]. GL and BL also enhanced in vitro PLB production in *Cymbidium dayanum* and *Cymbidium finlaysonianum* with the addition of chondroitin sulfate [108]. In *Gerbera jamesonii*, GL and RL illumination resulted in a highest number of axillary shoots and leaves number in the medium with 5 mg L^−1^ kinetin. However, in the same medium, a high fresh weight was obtained in WL [136].

On *Cymbidium* Waltz ‘cv Idol’, the highest shoot formation (80%) was observed in the medium containing 0.1 mg L^−1^ N- acetylglucosamine (NAG), under RL and 1 mg L^−1^ under GL; the fresh weight of PLBs was highest at 0.01 mg L^−1^ NAG under GL [100]. In the same orchid, six times of breaking the weekly light by 1 day of G-lighting during R-LED illumination showed optimal numbers and formation rates of PLBs. Optimal shoot formation was obtained by treatments of Fl+interval lighting of G-LED and B-LED+G interval lighting [95].

In combination with RL and BL, GL also positively affects plant growth, including leaf growth and early stem elongation [196,197], and is involved in the orientation of chloroplasts and in regulation of the stomatal opening [198].

In *Solanum tuberosum* plantlets in vitro, the addition of GL to the combined RL and BL increased stem diameter and leaf area, and the amounts of chlorophyll, soluble sugar, soluble protein and starch. The addition of GL to the combined RL and BL contributed to the growth and development of *Solanum tuberosum* plantlets more than the combined of RL and BL without GL [64].

Further research is necessary to understand the role of radiation oscillating around 550 nm, since the studies in this field are very limited and are mainly conducted in combination with other spectral wavelength radiations under in vivo conditions.

### 2.6. Yellow Light Effects on Shoot Proliferation and Plantlet Morphology

The reduction of apical dominance seems to be the main effect determined by YL and by the GL [128,135]. YL applied to cultures of *Prunus domestica* rootstocks Mr.S.2/5 and GF677 reduced apical dominance [199]; in *Malus domestica* rootstock M9, this light induced a production of axillary shoots greater than that detected under BL and FRL but still lower than that detected under RL [135]. Similar to the RL, the YL and GL induced a greater elongation of the internodes and outgrowth axillary shoots than the BL; in particular, the YL stimulated longer internodes in *Prunus domestica* rootstocks Mr.S.2/5 [142]. YL illumination induced higher proliferation in *Populus alba* × *P. berolinensis* [129].

YL irradiation followed by the RL one induced higher shoot proliferation (98%), a higher number of shoots per explants and early PLB formation, differentiation and shoot initiation in *Dendrobium sonia* [116]. YL elicited response of callus multiplication in *Vitis vinifera* [200]. YL also determined a higher leaf area and fresh weight and a lower shoot length in *Dendrobium sonia* [116]. YL showed a smaller increase in mean fresh weight as compared to BL but less than RL [135].

The YL positively affected growth in *Lactuca sativa* [201]. Based on current knowledge, the behavior of in vitro cultures subjected to YL would not be attributable to the actions of PHYs and BL photoreceptors.

### 2.7. Far Red Light Effects on Shoot Proliferation

Sunlight emits almost as much FR radiation as R radiation. Leaves absorb most RL but reflect or transmit most FRL [202]. As stated before, plants under a canopy or the lower leaves of plants spaced close together receive a greater proportion of FRL than RL radiation, i.e., a reduced RL:FRL ratio. Plants perceive this filtering of light and, in response, redirect growth and development according to the survival strategies of shade avoidance, increasing apical dominance and typically elongating in an attempt to capture available light [25]. In contrast, once sunlight has been reached, PHY and UVR8 inhibit shade avoidance. Several studies suggest that multiple plant photoreceptors converge on a shared signaling network to regulate responses to shade [203]. PHYs are the receptors of RL and FRL and are mainly involved in this perception, but plants shaded within a canopy also perceive reduced BL and possibly enriched green light through CRYs [190]. The detection of canopy gaps may be further facilitated by BL sensing phototropins and the UV-B photoreceptor, UVR8. Moreover, Zhen and van Iersel [204] reported that adding FRL consistently increased net photosynthesis of *Lactuca sativa* L. as compared to RL and BL. They attributed this effect to the increased quantum yield of photosystem II (ΦPSII).

The commonly applied Fl but also the R:B LEDs usually lack FRL, which is important for plant development, stem elongation and PHY activity, whereas they are abundant in GL and YL, which are less efficient for plants [35].

PHY in its active form, as may occur under high RL or RL:FR ratio, seems to alter the endogenous hormonal balance, reducing the apical dominance and increasing the shoot proliferation rate through enhancing lateral shoot development. On the contrary, low RL:FRL ratio or FRL alone reduces in vitro proliferation [2,205].

FRL appeared to increase node formation and decrease internode extension (but to a less degree than BL) as compared to the effects of RL. With dichromatic BL plus FRL, the effects on these two variables induced by BL were found to be slightly modified, indicating that the active form of PHY was only partially able to influence CRY-regulated physiological functions. While the effects of RL and BL and the RL:FRL effects during in vitro phases have been extensively examined, the effects of FRL alone have been less studied [59]. A high RL:FRL ratio or a low BL:RL ratio stimulated the sprouting of axillary buds in *Azorina vidalii* [74] and *Vaccinium corymbosum*, where, however, the presence of UV in the lighting device influenced shoot length differently in different cultivars [206]. Even in *Spirea nipponica*, shoot proliferation was greater when explants were exposed to combinations of high-ratio RL and FRL [124]. In a study on *Oncidium* [89], the best results on PLB formation were obtained under R+B+FR LEDs. This study also indicated that this combined radiation or RL:FRL radiation significantly enhanced leaf expansion, number of leaves and roots, chlorophyll contents and fresh and dry weight. The highest propagation ratios for *Chrysanthemum × morifolium*, *Heuchera × hybrida*, *Gerbera jamesonii* and *Lamprocapnos spectabilis* were reported under light emitted by RL- and FRL-abundant G2 LEDs [35]. The G2 spectrum was favorable in most of the species tested, probably because of the high GL:BL and RL:FRL ratios, which provide a higher portion of active PHYs [207].

Under a constant fraction of RL and BL, root number, length of roots and stems and fresh weight of the plantlets was related to the FRL component of the total PPFD in the *Chrysanthemum morifolium*. At the higher intensity of FRL tested (9 μmol m^−2^ s^−1^ of the total 43 μmol m^−2^ s^−1^ of PPFD), a reduction of the previous morphogenic characters was observed [174].

On the *Prunus domestica* rootstock GF655-2 cultured in vitro in the presence of BA, at a photon fluence rate of 20 µmol m^−1^ s^−1^, FRL irradiation significantly promoted shoot proliferation as compared to the dark [141]. At a lower photon fluence rate of 9 µmol m^−1^ s^−1^ the response was lower than the other lights and similar to that detected in the dark. Based on the data obtained in their experiments, the authors concluded that the proliferation rate induced under BL, FRL and WL strongly depended on the photon fluence rate, while no statistically significant differences could be found in the effects of RL irradiation at different photon fluence rates. In *Pyrus communis*, FRL was advantageous for shoot number, but shoot quality was inferior because of low shoot weight, hyperhydricity and chlorosis as indicated by the low total chlorophyll and carotenoid content [59]. Werbrouck et al. [94] reported the negative effect of FRL on in vitro biomass production of *F. benjamina* showing a reduction in the total number of shoots and in both shoot cluster and callus weight.

A reduced RL:FRL ratio (1:1.1) had an inhibitory effect on the growth of two *Calanthe* hybrids [184].

In microcuttings of a *Prunus cerasifera* rootstock, BL and WL produced a higher number of nodes, with shorter internodes compared to RL or FRL or dark. Differently, the proportion of nodes producing outgrowing of lateral shoots was higher in RL followed by FRL than in WL, BL or dark because of the weakening of apical dominance induced by the former two lights [125]. However, the highest proliferation of new shoots was seen in WL because of the high number of new nodes. Even here, as evidenced also by Baraldi et al. [141], the effectiveness of FRL required prolonged exposures and was dependent on photon fluence rates [125]. On M9 rootstock of *Malus domestica*, the development of phytomers appeared to be primarily caused by the active form of PHY, with a marginal effect from BL. Shoot growth, which combines internode elongation, development of the phytomer and branching, was highest under RL and the lowest under BL and FRL, showing the largely positive role of PHY photoequilibrium. FRL was the most inhibiting light type, reducing the proliferation rate compared with BL. Under FRL, reduced stem elongation was due to the very small number of phytomers formed [135].

## 3. Effects of Light Intensity

The selection of the optimal light intensity to support in vitro proliferation and growth is also important for an optimization of the processes. Among others, light intensity regulates the dimension of leaves and stems, as well as their morphogenic pathway, and is involved in pigment formation and hyperhydricity [208].

In vitro cultures are subjected to a much lower light intensity as compared to those grown under open field conditions. The permanent low light conditions in vitro have been considered a limiting factor for photosynthesis and for supporting plant morphogenesis in vitro, so it is necessary, in most cases, to supply sucrose to the medium [209]. In vitro plants are also very susceptible to high light conditions [210] and prone to photoinhibition [211]. Too high irradiation can severely damage the photosynthetic apparatus and photosynthetic pigment synthesis [48,212], leading to the formation of harmful free oxygen radicals and damage to cells [213].

In Table 3, the research that mainly addressed the effects of different light intensities is shown, but only in a few of the studies shoot proliferation is examined.

The optimal value of the PFD for plantlets changes from species to species and the predominant in vivo light conditions may give an indication of the requirements for optimal culture growth in vitro [75]. In *Alocasia amazonica* [222] and *Momordica grosvenori* [219], shoot length increased with the reduction in light intensity, an adaptation mechanism indicating that these species can survive in low light-intensity environments. In *Lippia gracilis*, the weight increase of plantlets grown under high light intensities indicates that this species originates in a semiarid environment where high irradiance (HI) incoming light occurs [119]. Evidence has been previously presented [178] that plants adapted to an environment with incoming HI present better photosynthetic rates and high growth rates under intense light. In an extensive study on the photosynthetic pigments, Lazzarini et al. [119] concluded that the increase in chlorophyll b content under low irradiance (LI) is indicated as an important marker of plant adaptation to shaded environments because this pigment is more efficient for capturing the photons of the higher wavelengths of the spectrum that are mainly present. Furthermore, it is worth noting that the type of explant also influences the amount of photosynthetic pigment: leaves of plantlets generated from apical explants had higher amounts of chlorophyll a, total chlorophyll and carotenoids regardless of light conditions, whereas the amount of chlorophyll b resulted in more plantlets generated from the lateral buds of nodal segments. Moreover, an increase in the synthesis of carotenoids was observed in plants grown under high light intensities and was associated with the photoprotection exerted by these pigments within the photosystems. In *Lippia gracilis*, this increase led to better efficiency of the photosynthetic activity and, hence, the higher production of dry weight observed under these conditions [119]. In three different species, *Disanthus cercidifolius*, *Rhododendron* cultivars and *Crataegus oxyacantha*, low levels of irradiance (11 µmol m^−2^ s^−1^) were optimal for in vitro growth, while higher irradiance determined a decrease in shoot development and leaf chlorophyll content in *Disanthus* and *Rhododendron* cultivars, which are shade-tolerant species in their natural habitat. Plantlets of *Crataegus* generated from in vivo plants adapted to higher levels of irradiance resulted in tolerance to a wide range of irradiances in vitro. Only shoot extension was inhibited at the highest levels tested, whereas leaf chlorophyll content was unaffected. These differences were attributed to a differential adaptation to light determined by the natural habitats of these plants and of the possible direct effect of irradiance upon plant growth regulators in the culture system [75]. Different effects of rising light intensity were observed in *Plectranthus amboinicus* grown in vitro. In this species, intensities below or above the optimum (69 μmol m^−2^ s^−1^) led to the lowest growth. In fact, photosynthesis was inefficient under low light intensity (26 μmol m^−2^ s^−1^) but increased light intensities led to reduced concentrations of a, b and total chlorophyll, and carotenoids and thus of growth [48]. In *Withania somnifera* and *Achillea millefolium*, the treatments with the highest light intensity (60 and 69 µmol m^−2^ s^−1^, respectively) showed the highest levels of photosynthetic pigments but not the highest growth. Alvarenga et al. [172] concluded that the significant increase observed in chlorophyll and carotenoids under high light conditions would indicate that these pigments have the photoprotective function, as assumed by Biswal et al. [223], since they may be inefficient in absorbing light and increasing photosynthetic efficiency. They also attributed the damage of excess light to the photosynthetic apparatus to the production of free radicals, which may degrade these pigments [45,213]. Kurilčik et al. [174] on *Chrysanthemum (Chrysanthemum morifolium*), noticed that the maximal PFD (85 μmol m^−2^ s^−1^) used in their experiment induces light abnormalities on the leaf surface. In ginger [224], the growth was restrained when the light reached 180 μmol m^−2^ s^−1^ and the chlorophyll content decreased as the light intensity increased.

However, a different sensibility to light intensity seems to affect proliferation rate and the plantlet growth, and in most cases lower plant intensities are required for proliferation.

Based on the observation of the examined papers for this review, in Figure 1, the light intensities were grouped in ranges and the frequency of their use is shown. From this study, it emerged that whatever the light spectrum, the most used light intensities range from 20 to 80 µmoles m^−2^ s^−1^ and the most used intensity for proliferation is 50 (µmoles m^−2^ s^−1^).

In *Rubus* spp, rising WL fluence rates from 0 to 81 µmol m^−2^ s^−1^ did not improve the organogenesis from cotyledons [225]. In *Vaccinium corymbosum*, exposure at rising intensities from 55 up to 210 µmol m^−2^ s^−1^ improved proliferation and rooting ratios only with short time applications (7 days). Longer exposure of the leaves (14 and 28 days) determined inhibition of growth and the red color of leaves and sprouts, and less vigorous plants after in vivo transferring [215].

However, a better multiplication under increasing irradiance, from 10 to 80 µmol m^−2^ s^−1^, resulted in *Pyrus communis* [218], in *L. gracilis* at 94 µmol m^−2^ s^−1^ [119] and in *Rosa hybrida* from 4 to 148 µmol m^−2^ s^−1^ [221]. In this last species, higher irradiance (66 and 148 µmol m^−2^ s^−1^) showed better effects on shoot proliferation, but leaf chlorosis was observed and better results on shoot growth were obtained at 17 µmol m^−2^ s^−1^ [221]. The chlorosis occurring at the higher levels of irradiance may be due to photochemical oxidation, photoinhibition or chloroplast damage [226].

In *Castanea sativa*, Sáez et al. [227] highlighted a correlation between light intensity and the addition of sugar to the growth medium. They demonstrated that HI (150 µmol m^−2^ s^−1^) and high sugar amounts (30 g L^−1^) produced an increase in photosynthetic activity and chlorophyll content and determined a higher proliferation rate and biomass production. However, a high proliferation rate was obtained even under LI with a higher sugar content in the medium. Thus, HI but also LI may be beneficial during the in vitro culture, but this is only possible in the presence of sucrose added to the culture medium.

Kozai [228], in *Cymbidium*, doubled in vitro growth by adding CO_2_ to the culture vessels at high PFD (230 μmol m^−2^ s^−1^), demonstrating that CO_2_ limitation may have a relevant role in enhancing the growth when high PFDs are adopted. The same was also true for *Actinidia deliciosa* where the proliferation rate and dry and fresh weight increased up to 120 μmol m^−2^ s^−1^ but decreased at higher rates. The biomass produced was also affected by light intensity, since both dry and fresh weight increased at the PPFD up to 120 µmol m^−2^ s^−1^, while only dry weight increases thereafter up to the highest value of 250 µmol m^−2^ s^−1^.

The photosynthetic rate was nearly four times higher when raising CO_2_ up to 1450 and 4500 μL L^−1^ compared to the lowest CO_2_ concentration tested (330 μL L^−1^) [220].

In fact, it has been shown that, just a few hours after the light was turned on, CO_2_ underwent a drastic reduction in concentration and sub-optimal CO_2_ availability has been correlated with reduced photosynthetic ability [229]. Thus, exogenous enrichments of this gas in the culture vessels improves photosynthesis at high PFDs [230,231].

Finally, most studies on the effects of light intensities have been carried out under Fl or W-LED. However, some studies revealed a relationship between the light spectrum and the intensity that affects plant growth and development. In the presence of BA, WL, BL and FRL, action on proliferation was dependent on the fluence rate [141].

Phytochrome has been shown to induce a high-irradiance response and low-irradiance response in *Prunus domestica* rootstock Mr.S. 2/5 [142]. Similar results were also obtained with the rootstock GF 677 in which the newly formed shoots were fewer but longer under the two intensities of RL (15 and 40 μmol m^−2^ s^−1^) than those treated with WL. In addition, the low intensity RL (15 μmol m^−2^ s^−1^) induced higher shoot multiplication as compared to the higher irradiance (40 μmol m^−2^ s^−1^). The formation of new shoots in the two species was affected differently by the increase in the RL irradiance, and shoot formation was found to increase in the cultures of Mr.S. 2/5 and decrease in those of GF 677. This result could be related to a species-specific response on which would depend different PHY regulation strategies [2].

## 4. Effects of Photoperiod

An organism’s life has evolved adaptation mechanisms that are related to environmental variations. Some of these variations exhibit regular cyclicality such as light:dark cycles, others fluctuate, such as temperature; however, all of them induce significant changes in the physiology and metabolism of most organisms, occurring in their life trajectory as characterized by the night and day cycle [232,233]. Plants possess the circadian clock, an endogenous time-keeping device that triggers and regulates physiological events in accordance with predicted daily changes in the environment. The input of light into the circadian clock is led by a set of photoreceptors such as the ZTL-type and UVR8 receptors [234]. Photosynthesis and stomatal movements are controlled by the circadian clock [235,236]. Among several physiological processes that include chromatin-regulation, diurnal rhythmic gene expression generates networks of genes that act specifically throughout the day or the night [237,238,239,240]. The circadian clock is an endogenous oscillator with a duration of approximately 24 h, and it is coordinated by external factors such as temperature and light. These external factors are relatively constant during the micropropagation procedure since there is no change in photoperiodism and thermoperiodism. During the shoot multiplication phase of in vitro cultures, photoperiod regimes of 16 h of light and 8h of dark are usually adopted. Plantlets in vitro are mixotrophic organisms, therefore nutrients such as carbohydrates are absorbed from the medium. In plantlets exposed to a 16:8 h photoperiod, the photosynthetic activity is intense at the onset of the light cycle and decreases rapidly thereafter. The block of CO_2_ assimilation depends on the rapid and progressive lower concentration of CO_2_ in the culture vessels. The CO_2_ availably in the culture vessels is largely generated during the respiration of sucrose supplied with the growth medium, since the gas exchange between the inside and the outside the vessel is almost absent (Abbot and [220,230,241,242]. The modification of the photoperiodic regime from a 16 h photoperiod cycle to a 4 h photoperiod cycle promoted the increment of fresh and dry weights of shoot clusters, and the number of neo-formed shoots from initial shoot explants in two *Prunus persica* rootstocks [243]. An analogous response was found in the *Prunus persica* cultivars Suncrest, Belle of Georgia and Evergreen when cultured in the presence of 10µM of BA in the medium [244]. However, Morini et al. [245] have found that the photosynthetic activity was only extended until 4 h after the beginning of the illumination, although the concentration of CO_2_, (under the 16/8 h regime) was not a limiting factor since at the end of the light period its availability was still much higher than that outside the vessel. From the same authors, the reduction of photosynthetic capacity was attributed to a reduced efficiency of the chloroplasts coupled with the lengthening of the light period. The promotive role of the 4 h photoperiod cycle on the shoot proliferation rate was hypothesized to be dependent on the diverse regime of photo-equilibrium of photoreceptors that promoted the reduction in apical dominance and development of axillary buds.

However, in studies carried out on other species, subjected to a 16 h photoperiod, low concentration of CO_2_ into the vessels was observed: *Pfaffia glomerata* [246], Solanum tuberosum [247,248], *Carica papaya* [249], *Castanea sativa* [227], *Vitis vinifera* [250,251], *Fragaria x ananassa* [252], *Hyptis marrubioides* and *Hancornia speciosa* [253].

## 5. Light and Plant Growth Regulators

Some in vitro studies highlighted the effects of light spectra on the effectiveness of endogenous- and exogenous-applied growth regulators.

### 5.1. Light Effects on Endogenous Growth Regulators

Endogenous auxins and CKs are the most involved growth regulators in regulating apical dominance [254]. Apical dominance and its correlative inhibition are determined by the synthesis of auxins by the apex [255]. In the classical model, it is hypothesized that these hormones are synthesized by the apex and transported downward into axillary buds, with subsequent direct downregulation of outgrowth, or indirect regulation via other mechanisms such as nutrient diversion, expression of genes that control the growth of axillary buds, adjustment of the auxin/cytokinin ratio, including activation of strigolactones capable of modifying the hormonal balance, and the apical dominance [256,257]. On the other side, an increase of CK quantity in tissues leads to a marked growth of axillary buds, counteracting the action of auxins. Studies on transgenic plants have shown that regulation of apical dominance by plant hormones is not determined by the absolute concentration of hormones but by the ratio between them [258]. In vitro shoot proliferation is strongly dependent on the ability of CKs to counteract apical dominance, i.e., the physiological control exerted by the apex over the induction and development of the new lateral meristems in axillary buds along the axis of the growing explant. Light acts mainly as a morphogenic signal in the triggering of bud outgrowth and initial steps in the light signaling pathway induce changes in the levels of cytokinin-like substances [259,260,261]. The effect of light in modulating endogenous CKs levels is well-known and has been demonstrated in several species such as *Rosa hybrida* and *Chlorella minutissima* (*Chlorophyta: Trenouxiophyceae*) [262,263]. In *Rosa hybrida*, in dark, inhibition of bud outgrowth is suppressed solely by the application of CKs. In contrast, application of sugars has a limited effect. Exposure of plants to WL induces a rapid (after 3–6 h) up-regulation of *RhIPT3* and *RhIPT5* genes involved in CK synthesis, of the *RhLOG8* gene involved in CK activation and of the *RhPUP5* gene involved in CK putative transporter and induces the repression of the *RhCKX1* gene involved in CK degradation in the node. This leads to the accumulation of CKs in the node and to the triggering of bud outgrowth [263]. In *C. minutissima* [262], a rise in endogenous auxin and CK and a decrease over time in gibberellin concentrations was observed in the actively growing cultures under light:dark conditions (L:D) and continuous dark+glucose (CD+G) but no increase was determined under continuous dark (CD). The L:D cultures had the largest CK increase.

It has been known for several years [264] that the bands of the light spectrum that have been shown to promote morphogenetic processes through the activation of the various photoreceptors are mainly represented by RL, FRL and BL. As stated in paragraph 2.1, RL increases the quantity of cytokinin in tissue, counteracting the action of auxins and thus determining an increase in the development of lateral shoots [139,140]. RL also regulates the synthesis of carotenoids and strigolactones [265]. Previous studies reported that RL decreased the IAA concentrations in maize epidermal cells [266].

The interaction between CK and PHY would induce, in the latter, an extension of the active form (P_fr_) even in conditions of dark and FRL [2]. In addition, other plant hormones may be modulated by light and by phytochrome directly. Among these are gibberellins [65] and brassinosteroids [267], another important category of growth regulators affecting cell elongation and cell division. Thus, RL may promote stem growth by regulating the biosynthesis of gibberellin or induce the expression of an auxin inhibitor gene to promote stem and root lengthening in grape [8]. In contrast, BL seems to affect more the auxin content (indoleacetic acid-IAA in particular). In fact, it was demonstrated that BL induced higher IAA content than RL in the leaves of the balloon flower [158] and thus it is more effective in promoting leaf growth. Significantly higher IAA contents occurred in the leaves under the BL:RL = 3:1 and BL:RL 1:3 and induced larger leaf areas compared to RL. Thus, BL appeared more beneficial for increasing IAA concentrations and for promoting better leaf growth than RL. However, in tobacco, a species in which BL stimulated shoot proliferation, contrasting effects of BL have been reported, since it was hypothesized that at higher intensities it determines the photoinactivation of IAA [67]. These mechanisms, both related to apical dominance and bud dormancy, are masked by WL, a condition under which cryptochrome and phytochrome are activated.

### 5.2. Effects of Light on Exogenous Applied Growth Regulators

In *Prunus domestica subsp. insititia*, clone GF655-2, BA, a promotive effect on proliferation was repressed under dark, whereas no proliferation was observed under light conditions without BA. It is noteworthy that at the highest BA supplied, the proliferation rate increased under the broadband WL, whereas it decreased under the monochromatic sources RL, BL and FRL [141]. Light and BA also proved to be indispensable factors in adventitious shoot formation from *Pinus radiata* cotyledons [268]. In *Spirea nipponica*, the interaction between CKs (0.25 mg L^−1^) and RL resulted in an enhancement of the shoot proliferation rate [123]. The same indications on the interaction between light quality and CK content were obtained on multiplication and growth during in vitro culture of *Myrtus communis* L. [120] and *Spirea nipponica* [124]. The highest number of shoots was obtained under RL or R:FR-LEDs with the higher CK concentrations tested in the media (5 µg L^−1^ i.e 1.1 and 0.5 mg L^−1^, respectively).

At lower BA levels (0.4 mg L^−1^), 4 weeks of RL:FRL at low fluence followed by 1 week of WL at higher fluence rate produced almost the same proliferation levels and optimal growth [124]. If the CK concentration was below the optimal level, the production of axillary shoots was greater in the RL; at higher CK concentration, the multiplication rate decreased [2]. The effect of light spectrum differs, however, in relation to the concentration of CK applied: at the optimal concentration, WL provided responses better than those obtained with RL and BL. Thus, the quantity of applied CKs may decrease under RL. Analogously, CK incorporation into the culture medium annulled the promoting effect of RL in axillary bud proliferation from azalea apices and adventitious bud regeneration from *Petunia* spp. leaf segments [269,270]. Probably, light quality and hormone application may affect the morphogenesis of in vitro plants, in part because of changes in sink strength and, as a consequence, to redistribution of active growth [71].

Panizza et al. [72] analyzed the effect of spectral composition on axillary proliferation of lavandin *(Lavandula officinalis* Chaix • *L. latifolia* ViUars cv. Grosso) in relation to the application of exogenous BA, putrescine (Put) and endogenous ethylene production. The effect of BA was predominant over the light quality, whereas in BA-free medium, shoot number was enhanced under BL, WL and RL at low photon fluence rates. BA, however, could reduce the inhibiting effect of BL and UVL at high photon fluence rates. Exogenous Put stimulated axillary bud proliferation under some light treatments in the presence of BA, although the short fluence RL treatment was critical to allow the positive effect of Put on shoot formation. A positive correlation between biotic ethylene production and shoot formation was evidenced under FRL at a high photon fluence rate in the presence of BA. In the BA-free medium, further evidence of the correlation between biotic ethylene and the proliferation process was given since the biotic emanation increased under those radiation treatments (RL, BL and WL), which also improved shoot number. The authors conclude that in the evaluation of the responsiveness of a tissue to radiation in vitro, great care should also be devoted to radiation-induced changes in the abiotic environment (e.g., ethylene release) [72].

## 6. Discussion and Conclusions

Several papers on different species concern the effects of light on in vitro proliferation to assess the light properties capable of enhancing the efficiency of the micropropagation process, also ensuring consistent energy savings, as compared to traditionally used Fls lamps, or the broad range of WL. However, the results are often conflicting. Many authors ascribe these results to the different responses to light of plant species, cultivars or even explant types [119], plant stage development [122], medium composition [143] and micro-environmental characteristics such as PPFD [174] and vessel ventilation [146]. However, a large cause of variability may be tied also to the difficulty in applying uniform intensities along the shelves, and/or the use of the right spectral composition for each light quality.

Moreover, the lack of sufficient in vitro experimental protocols like those available for in vivo study, which would make the effects of light clearer, limits the comparability of the experiments [34]. The issues of major concern, among others, in this regard are (i) the short timescale in which these experiments are carried out (mostly a propagation cycle), (ii) the quality and quantity of exogenous applied growth regulators, (iii) the narrow range of light intensity values within which the efficiency of axillary multiplication of explants occurs and (iv) the mixotrophic state of plantlets. Concerning the first issue, the short-time experiments strongly limit the comprehension of the effects of light spectra on the stability of proliferation and plantlet growth during subsequent multiplication cycles (see particularly the RL effects). Concerning the second one, due to the interaction of light with endogenous growth regulators (particularly CKs), attention must be paid to the doses of the exogenous growth regulators applied. It seems from the examined literature, in fact, that RL effects are visible under low CK concentrations in the medium, whereas WL effects are even visible under high CKs doses [83]. Too high CKs quantities mask the effects of RL or may determine growth alteration. Concerning light intensities, excessive LIs or HIs may determine low growth rates, photoinhibition and may mask light spectra effects. Moreover, information on how the mixotrophic metabolism of a plantlet grown in vitro affects the development and morphology of the microcutting is scarce.

In this review, several research are presented regarding the different response of species and cultivars to different light spectra, intensities and photoperiod and it seems that some general indications arise from the different studies. Concerning the optimal irradiance intensity, it has been hypothesized that the prevailing light conditions under the natural habitats of some species can be used to indicate their requirements for optimal in vitro growth [75]. Evidence have been presented that plants adapted to an environment characterized by high light intensities present better photosynthetic rates and high growth rates under in vitro intense light, whereas shade-tolerant plants are damaged by high intensities. A survey of the tested literature revealed that in most species, whatever the light spectrum, the most used light intensities range from 20 to 80 µmoles m^−2^ s^−1^ and the most used intensity for proliferation is 50 µmoles m^−2^ s^−1^. Better growth, however, have been registered especially in plants adapted to high intensities (see *Saccharum officinarum*, *Actinidia deliciosa*, *Lippia gracilis*, etc.) at intensities up to or exceeding 80 µmoles m^−2^ s^−1^. Significant improvements on in vitro fresh and dry weights of shoot clusters, and the number of neo-formed shoots from initial shoot explants were obtained, also modifying the photoperiodic regime from a 16 h photoperiod to a 4 h photoperiod cycle, thus permitting the plantlets to replace the CO_2_ [243,244]. In fact, in plantlets exposed to the 16:8 h photoperiod, the photosynthetic activity is intense at the onset of the light cycle and decrease rapidly thereafter because of the rapid and progressive lower concentration of CO_2_ in the culture vessels. Moreover, the promotive role of the 4 h photoperiod cycle on the shoot proliferation rate was hypothesized to be dependent on the diverse regime of photo-equilibrium of photoreceptors that promoted the reduction in apical dominance and development of axillary buds [243]. In this view, also adding CO_2_ [220] or aerating the vessels [146] proved to be effective in enhancing in vitro growth.

Concerning light spectra, RL alone or high RL:FRL ratios seem to enhance shoot proliferation, as well as PLB and callus formation, in many species. The main effects of RL are tied to the promotive role of phytochrome in the synthesis of CK in tissue, which counteracts the actions of auxins, increasing the development of lateral shoots. RL also regulates the synthesis of carotenoids and, in particular, strigolactones that seem to regulate apical dominance by modification of auxin fluxes [271]. The stimulatory effects of RL seem to be exerted during the beginning of the multiplication phases. However, different reports indicated that RL alone is not able to activate the pathway of chlorophyll synthesis and may determine excessive stem elongation and leaf disorders, the so-called Red Light Syndrome [36]. In fact, when plants are grown under 100% monochromatic RL a strong decrease in photosynthetic capacity, rates of electron transport, dark-adapted Fv/Fm and leaf thickness, as well as unresponsive stomata and reduced leaf pigmentation occurs [272]. BL is effective in increasing callus formation and the number of axillary buds but exerts an inhibitory action on buds sprouting (increase in apical dominance). It has been demonstrated that this light mostly controls some morphological characteristics such as shoot length and enhances chlorophyll synthesis and chloroplast development. RL, on the other hand, would remove the apical dominance but seem to reduce the formation of new axillary buds. Hence, a minimum threshold of BL is necessary for normal plant growth [146]. Moreover, regulating the spectral quality particularly by the BL improves the antioxidant defense line and is directly correlated with the enhancement of phytochemicals [65,90,166] or with the regulation of gene expression [167]. All these reasons would explain why the RL:BL illumination resulted effectively in a wide range of species. Moreover, more recently, an abundance of evidence has indicated the role of GL in carrying information about the environment to the plants, because it is involved in the shade avoidance response, but also in regulating different biological, morphological and biological processes in vitro and in vivo [189]. The addition of GL to the combined RL and BL contributed to the proliferation, the growth and development of some in vitro cultures. In a few cases, even the addition of YL seems to improve plant proliferation and growth. In addition, the absence of ultraviolet light may determine foliar intumescence and could become a serious limitation for some crops lighted solely by narrow-band LEDs [273]. Thus, the use of monochromatic or combined R- or B-LEDS may determine a mismatch with the photosynthetic spectrum. The application of the broad band WL may overcome this problem [44]. In some species, better results have been obtained under W-LEDs [109,112,130]. Even if WL is not as effective as RL in overcoming apical dominance, high proliferation rates are obtained when CKs are added to the medium. In most cases, the best propagation was obtained at higher CK ratio [141]. It seems that the CK ratio may be enhanced in woody species under WL to obtain high and stable proliferation. However, in some species, after long-time cultivation under WL the rate of newly formed sprouts was reduced regardless of the CK concentration but increased when RL was applied to the crops [2]. Thus, in some cases, an early phase of RL irradiation of at least 2 weeks [122], followed by growth under a WL, may be advisable. The use of an initial stimulatory effect of RL or RL enriched followed by the WL may also improve proliferation and somatogenesis [126] in species that are particularly difficult to regenerate in vitro and/or with an high sensibility to higher concentration of CKs in the medium, such as *Euphorbia milii* and *Ceratonia siliqua* L. (Cavallaro et al., unpublished data). Moreover, the exposition to a period of RL:FRL followed by the WL may enable a reduction in exogenous growth regulator concentrations, mainly CKs added to the medium [124], which may be unnaturally high in vitro. This reduction may be favorable for enhancing the following phases of the in vitro process (rooting and acclimation). Finally, currently, lamps with a more optimal spectral composition of WL enriched in the most useful wavelengths (BL, RL and GL) are already available on the market [185,274] for vertical farming systems and could be interesting for in vitro production after appropriate investigation.

## Figures and Tables

**Figure 1 plants-11-00844-f001:**
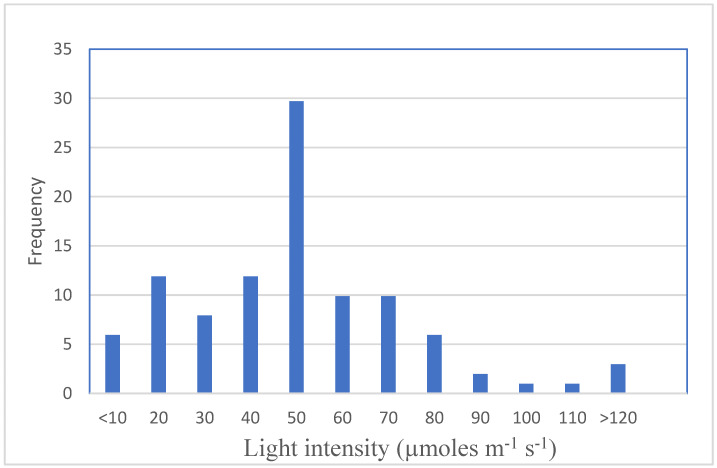
Frequency of light intensities used in literature for proliferation.

**Table 1 plants-11-00844-t001:** Summary of the use of LED lighting on in vitro propagation of herbaceous and shrub species.

Studied Species/Explant Type	Light Intensity and Photoperiod	Light Spectra	Growth Regulators in Medium	Results on In Vitro Proliferation	Morphogenetic Response	Authors and Year
*Nicotiana tabacum* L. var. Wisconsin 38)/Callus	mW cm^−2^: 0, 0.0028; 0.024; 0.13; 0.37; 0.60; 0.80 photoperiod 16 h	8 narrow band lights: 371, 419.5, 467, 504, 550, 590, 660, 750 nm, 4 commercial broad band-Fl lamps	For shoot differentiation: 2 mg L^−1^ K, 2 mg L^−1^ IAA, 80 mg L^−1^ adenine sulfate dihydrate	Near UV at low intensity (0.024 mw/cm^2^) and BL at higher intensities, callus growth and shoot initiation.	Higher carotenoids, porphyrins, associated with the high irradiance response.	[67] No LEDs
*Vitis vinifera* L. hybrid ‘Remaily Seedless’/Node shoots (axillary bud proliferation)	µW cm^−2^: 1500 for RL 1600 for BL light	RL BL No LED	BAP at 5 µM	BL = more shoots in the medium containing the lower concentration of manganese sulphate.	BL = larger shoots and more vigorous plantlets.	[68] No LEDs
*Saintpaulia ionantha* Wendl cv. Sona/leaves and *Lycopersicon esculentum* Mill./Cotyledons cv. UC 105		Continuous light and daily light pulses		RL ad WL = highest bud regeneration in *L. esculentum*, BL in *S. ionantha*		[69] No LEDs
*Vitis vinifera* L. hybrid ‘Remaily Seedless’/Leaf axillary buds	10-h and 16-h photoperiods	WL of various spectral irradiances, BL and RL light.	Apex removal from the explant was evaluated.	BL = best for shoot production. Under W, shoot production was greater with ratios of BL:RL of 0.6 to 0.9.		[70] No LEDs
*Solanum tuberosum* L., cv. Miranda/Three- to four-node shoots (15 mm)	160 µmol m^−2^ s^−1^ 18 h (LD) or 10 h (SD). photoperiod	RL, BL	With or without 1 mg L^−1^ IAA or 1 mg L^−1^ K.	BL and K = better tuber production. RL and IAA application = high root/shoot ratio. Darkening strongly promoted tuber formation	Under BL, K increased total fresh weight and root (>stolons)/shoot ratio).	[71] No LEDs
Lavandin (*Lavandula officinalis* Chaix ×*Lavandula latifolia* Villars cv. Grosso)/Node explants	µmol m^−2^ s^−1^: Fl high fluence (HF) = 66 Fl low fluence (LF) = 7 RL (HF) = 7 RL (LF) = 1 FrL (HF) = 8 FrL (LF) = 2 BL (HF) = 13 BL (LF) = 1.5 UVL (HF) = 62 UVL (LF) = 5	D control WL RL Fr L FrD (25 min Frh + 30 d D) FrRD (25 min Frh + 10 min R high + 30 d D) BL UV (UV A and B)	BA (l µM), putrescine (Put, 1 and 10 µM)	Low fluence RL = higher shoot number in presence or absence of BA. At low fluence rates also WL and BL enhanced shoot number on BA-free medium. 10 µM putrescine + Ba improved proliferation.	Rl and D positively affected shoot length.	[72] No LEDs
*Begonia gracilis* Kunth/Direct somatic embryogenesis from petiole explants.	45 µmol m^−2^ s^−l^	RL and D	0.5 mg L^−1^ kinetin	Somatic embryo production was higher under RL that in the dark.		[73] two cycles
*Azorina vidalii* (Wats.) Feer (Dwarf shrub)	50 µmol m^−2^ s^−1^: 16 h photoperiod	High and low ratios of BL + RL (2.3; 0.9) or RL + FRL (1.1; 0.6). Control: Fl	in vitro shoots no growth regulators	High ratio of RL/FRL light or BL/RL = the highest number of axillary shoots as compared to control.	Low ratio RL/FRL = maximum plant length and leaf area	[74] three months
*Rhododendron* spp./Axillary buds *Disanthus cercidifolius* Maxim./Shoot. *Crataegus oxyacantha* L./Axillary bud	µmol m^−2^ s^−1^: 11, 25, 55, 106 and 161 for *Disanthus* and *Crataegus*; 16, 26, 60 and 120 for Rhododendron	RL, GL and BL	*Rhododendron* 2.5 µM 2iP. *Disanthus cercidifolius* 3 µM BAP *Crataegus oxyacantha* 2.5 µM BAP and 0.5 µM IBA.	RL promoted axillary branching. All cultures grew well at low levels of irradiance	RL promoted shoot extension.	[75] No LEDS
*Solanum lycopersicum* cv. UC 105 an aurea (au) mutant and its isogenic wild type/Organogenesis from hypocotyls	µmol m^−2^ s^−1^: Fl = 50 0, 2.5 and 5 the other light treatments. 16 h photoperiod	D and Fl for aseptic seed germination RL, FRL, BL for regeneration.	Hormone free medium	All genotypes germinated under Fl. The wild type even under dark. Under RL, FRL and BL, hypocotyls showed a position-dependent regeneration.		[76] two cycles No LEDs
*Petunia* x *atkinsiana* ‘Surfinia White’ cv.‘Revolution’/ Leaf explants	19–21 µmol m^−2^ s^−1^	WL, RL, BL, GL	0.1 mg L^−1^ NAA, 1 mg L^−1^ BAP	Organogenesis was carried out in darkness. WL, GL and RL = the highest number of adventitious shoots.	Blue = the longest shoots and the biggest leaf area.	[77]
*Lilium oriental* hybrid ‘Pesaro’/In vitro-raised bulbs	70 mmol m^−2^ s^−1^ 12 h photoperiod	D, Fl, RL, BL, RL + BL (1:1).	1.0 mg L^−1^ BA + 0.3 mg L^−1^ NAA	Fl, BL, and BL + Rl enhanced, plant regeneration as compared to D.	Bulblets under R + B were bigger in size, in fresh and dry weight.	[78]
*Begonia erythrophylla* J. Neuman/Petiole explants.	μmol m^−2^ s^−1^: WL, RL, and BL, and RL + BL = 35 Fr = 5 Continuous light	D, WL, R, B, RL + BL (1:1), FR	0.54 mM NAA, 4.44 mM BA	RL or WL, as pre-treatments, promoted competence. RL or WL during culture, enhanced shoot number.	White light produced best developed and expanded shoots.	[79] No LEDS
*Cymbidium* *Twilight Moon* cv.‘Day Light’/ PLB segments.	45 μmol m^−2^ s^−1^ 16-h photoperiod	RL, RL + BL (3:1), RL + BL (50:50), RL + BL (1:3), BL. Control = Fl (PGF)	For callus induction from PLBs: 0.1 mg L^−1^ NAA and 0.01 mg L^−1^ TDZ For callus proliferation: 0.1 mg L^−1^ NAA and 0.01 mg L^−1^ TDZ. For PLBs production from callus: no growth regulators.	RL determined more callus induction; RL + BL (3:1) and PGF more callus proliferation RL + BL (1:3) more PLBs formation		[80]
*Lactuca sativa* L./Cotyledon explants	35 μmol m^−2^ s^−1^	D, WL, RL, BL, BL + RL	0.44mM BA, 0.54mM NAA	Light improved organogenesis as compared to D. RL and WL light promoted shoot production.		[81] No LEDS
*Fragaria × ananassa* Duch. cv. Toyonoka/Leaf discs	2000 lux	GL, RL, BL and YL Fl as control	1.5 mg L^−1^ TDZ and 0.4 mg L^−1^ IBA.	Red and Green films determined the highest percentage of shoot regeneration and the max number of shoots per explant	RL and GL = a lower chlorophyll a/b ratio and higher antioxidant enzymes activity.	[82] No LEDs
*Euphorbia milii* *Des Moul.*/Inflorescences *Spathiphyllum cannifolium (Dryand. ex Sims) Schott*/In vitro shoots	μmol m^−2^ s^−1^: 50 for *Euphorbia*: 35 for *Spatifillum* 16 h photoperiod	LEDS: RL, BL, RL + BL (1:1); BL + FrL (1:1); RL + FrL (1:1) Fl = Control	For *E. miliii* 1 mg L^−1^ BA, and 0.3 mg L^−1^ IBA. For *S. cannifolium* 3 mg L^−1^ BA, and 1 mg L^−1^ IBA.	*S. cannifolium =* best shoot proliferation under RL, RL + FRL.	For *E.milii.* BL = higher fresh and dry weight, and leaf number. For *Spatifillum.* BL= the highest chlorophyll and carotenoid contents. In both species, RL= higher plantlet length and higher fresh and dry weights.	[83]
Two species of Petunia: *Petunia × atkinsiana* *(Sweet) D. Don* and *P. axillaris* (Lam.)/Leaf tissue	50 µmol m^−2^ s^−1^ 16-h photoperiod	Fl, D	5.7 μM IAA and 2.25 μM Zeatin.	*Petunia × atkinsiana* did not regenerate in darkness. Both species regenerate under light.		[84]
*Vitis vinifera* L. cvs: Hybrid Franc, Ryuukyuuganebu (a wild grape native to Japan) and Kadainou R-1/Nodal segments	50 µmol m^−2^ s^−1^ 16-h photoperiod	RL and BL PGF light was used as control	PGR-free medium	No differences or slight differences on proliferation due to light treatments	RL = longest shoots. BL = higher chlorophyll content, leaf and stomata number per explant.	[85]
*Phalaenopsis* hybrid cv. Cassandra Rose/PLBs from in vitro germinated seeds and flower-stalk nodes.		RL, RL + BL (9:1, 8:2), RL + WL (1:1) Fl		RL + BL (8:2) = the highest PLBs development. RL + BL (9:1) = the highest shoots number. Shoot tips had higher PLBs induction under RL and BL.	RL and BL =the highest PLBs fresh weight. LED lights = more fresh weight, Height and leaf length.	[86]
*Oncidium* Sweets Sugar/Shoot apex		Fl (control), RL, BL		RL promoted PLB induction from shoot apex with the highest proliferation rate; BL the highest differentiation.	RL determined the highest content of carbohydrates. BL the highest protein content and enzyme activity.	[87]
*Cymbidium finlaysonianum Lindl.*/PLBs	16 h photoperiod	RL, Fl.		RL increased PLBs proliferation and number		[88] No LEDs
*OncidiumGower Ramsey*/Embryogenic calli	50 µmol m^−2^ s^−1^	D, Fl, BL, RL or RL + BL + Fr (RBFr)	0.1 mg L^−1^ NAA and 0.4 mg L^−1^ BA	PLB formation and plantlet conversion was higher under (RBFr) LEDs and Fl.	RBFr enhanced leaf number and expansion, root, chlor. contents, fresh and dry weight.	[89]
*Oncidium Gower Ramsey*/Shoot tips	11 µmol m^−2^ s^−1^	Fl(control)RL, BL, YL and GL.	For PLBs induction, 1.0 mg L^−1^ BA, For PLB proliferation: 1.0 mg L^−1^ BA, 0.5 mg L^−1^ NAA.	RL enhanced PLB induction and multiplication, but low differentiation BL promoted PLbs differentiation into shoots	RL = the highest PLBs fresh weight and starch content. BL = higher chlorophyll, carotenoids and soluble protein content.	[90]
*Cymbidium finlaysonianum Lindl.*, *Cymbidium Waltz* cv.‘Idol’, and *Phalaenopsis* cv:‘1327’/protocorm-like bodies (PLBs)		RL, BL and YL fluorescent films		RL and YL increased the number of PLBs of *C. Waltz.*, RL, BL and YL increased the formation of shoots. RL and BL increased PLBs number in *Phalaenopsis*.	RL, BL and YL increased the fresh weight of PLBs in *C.finlaysonianum.*	[91] No LEDS
*Dendrobium officinale Kimura & Migo*/PLBs	70 µmol m^−2^ s^−1^ 16 h photoperiod	D, Fl, RL, BL; RL + BL (1:1); RL + BL (2:1); and RL + BL (1:2).	0.5 g L^−1^ NAA, 0.2 g L^−1^, 6-BA	BL, RL + BL (1:1) and RL + BL (1:2) = higher percentage of PLBs producing shoots and the number of shoots produced per PLB	BL and different RL + BL ratios enhanced chlorophyll and carotenoids. BL, Fl, and RL + BL (1:2) produced higher dry matter.	[92] three cycles
*Cymbidium insigne Rolfe*/PLBs		WL, RL, BL and GL	Chondroitin sulfate The medium was added with Chitosan H or hyaluronic acid (HA9)	GL and 0.1 (mg L^−1^) and Chitosan H determined the highest PLBs and shoot formation.	Fresh weight of PLBs was higher at HA9 (1 mg L^−1^) treatment with GL.	[93]
*Ficus benjamina* L. cv Exotica		BL, RL and FR. Fl as control	0.5 mg L^−1^ IAA and 2 mg L^−1^ BA.	BL increased shoot number, and callus growth.	RL determined an increase in shoot length.	[94]
*Cymbidium Waltz* cv ‘Idol’/5 mm protocorm-like bodies (PLBs)	50 μmol m^−2^ s^−1^ 16 h photoperiod	Fl, RL, BL, GL, Fl + GL, RL + GL, BL + GL. The last three treatment were subjected to 1d green exposure every 7d.	No growth regulators	RL + GL and BL promoted the highest PLB formation. Fl + GL and increased shoot formation from PLBs.	Fl gave the highest fresh weight. B + G the highest SOD activity.	[95]
*Brassica napus* L. cv Westar/Cotyledons from germinated seeds.	60 μmol m^−2^ s^−1^ 12 h photoperiod	Fl, BL, BL + RL (B:R = 3:1, 1:1, 1:3) RL.	For induction: 2,4-D in the dark; for shoots differentiation: 0.8 mg L^−1^ BA, 0.5 mg L^−1^ NAA; for shoots proliferation 1.0 mg L^−1^ BA.	The proliferation rate was greater under BL and BL:RL = 3:1 than under Fl	BL:RL (3:1) = higher fresh dry mass, chlorophyll a, soluble sugar, stem diameter, leaf stomata surface, than under Fl. Starch was higher in plantlets cultured under R light as compared to Fl.	[51]
*Linum usitatissimum* L., cv. ‘Szafir/Hypocotyls	50 µmol m^−2^ s^−1^	Light (Fl) or D conditions	0.05 mg L^−1^ 2,4-D and 1 mg L^−1^ BA	Shoot multiplication was about twice higher in light-grown cultures than those in darkness.	Fresh and dry mass and cyanogenic potential of light-grown cultures was about twice higher than those in the dark	[96] two cycles
*Solanum tuberosum* L. cvs Agrie Dzeltenie, Maret, Bintje, Désirée and Anti/Shoot tips from in vitro plantets	40 µmol m^−2^ s^−1^	Fl, warm WL light BL, RL, RL + BL (9:1 RB) and RL + BL + FR (70:10:20 RBF)	0.5 mg L^−1^ zeatin riboside, 0.2 mg L^−1^, GA_3_ and 0.5 mg L^−l^ IAA.	RL + BL (9:1) doubled the regeneration percentage of all cultivars after cryoconservation		[97]
*Abeliophyllum distichum Nakai*,/Apical and axillary buds	40 µmol m^−2^ s^−1^	BL, RL + BL (1:1 RB), RL, Fl	BA 1.0 mg L^−1^, IBA 0.5 mg L^−1^	BL and RL + BL promoted shoot proliferation.	RL increased shoot length.	[98]
*Dendrobium kingianum* *Bidwill ex Lindl.*/PLBs	50 μmol m^−2^ s^−1^ 16 h photoperiod	RL, BL, RL + BL (1:1), GL and WL, Fl = control	MS medium supplemented with 412.5 mg/L ammonium nitrate, 950 mg/L potassium nitrate	BL and RL determined the highest PLBs number. RL and WL increased the percentage of shoot formation.	BL increased chlorophyll percentage, RL determined the highest fresh weight.	[99]
*Cymbidium Waltz* cv ‘Idol’	16 h photoperiod	GL, RL, BL	N- acetylglucosamine (NAG) 0, 0.01, 0.1, 1, and 10 mg L^−1^	GL and RL + NAG determined the highest PLB formation rate RL or GL + NAG determined high shoot formation (80%)	Fresh weight of PLBs was highest at 0.01 mg L^−1^ NAG under green LED	[100]
*Saccharum officinarum* L., variety RB92579/in vitro grown plantlets	µmol m^−2^ s^−1^: (1) 72 (2) 60 (3) 57 (4) 53 (5) 77 16 h photoperiod	(1) BL + RL (70:30) (2) BL + RL (50:50) (3) BL + RL (40:60) (4) BL + RL (30:70) (5) WL	1.3 µM BAP.	BL + RL (70:30) gave the highest multiplication followed by 50:50. WL the lowest one.	BL + RL (70:30) and (50:50) = the highest total fresh weight. WL = the highest total chlorophyll content	[101]
*Scrophularia takesimensis* Nakai/Leaf, petiole, and stem explants	45 µmol m^−2^ s^−1^ 16 h photoperiod	Fl, RL, BL	2.0 mg L^−1^ BA and 1.0 mg L^−1^ IAA	Fl = the highest number of shoots per leaf, petiole and stem explants	RL gave better shoot growth followed by Fl and BL.	[102]
*Curculigo orchioides* Gaertn./Leaf explants	60 µmol m^−2^ s^−1^	BL, RL, RL + BL (1:1). Fl as control.	4 mg L^−1^ BA	BL determined the highest percentage of shoot organogenesis and shoot buds per explant.		[103]
*Fragaria x ananassa* Duch. cv. ‘*Camarosa*’/Encapsulated shoot tips	50 μmol m^−2^ s^−1^ 16 h photoperiod	Fl (control) RL + BL (9:1 R9B1); RL + BL (7:3 R7B3); RL + BL (1:1 R5B5); RL + BL (3:7 R3B7);	Hormone free medium for plantlets development, and 4.9 µM IBA or 6.7 µM BA plus 2.3 µM K for shoots proliferation	RL + BL (1:9) were most effective *for* in vitro sprouting of encapsulated strawberry shoot tips.	R7B3 promoted shoot length, chlorophyll content, fresh and dry biomass accumulation.	[104]
*Panax vietnamensis Ha et Grushv*/Callus	20–25 µmol m^−2^ s^−1^ 16 h photoperiod	D, Fl, BL, GL, YL, RL, WL, and RL + BL: 90:10, 80:20, 70:30, 60:40, 50:50, 40:60, 30:70, 20:80, 10:90.	For embryogenic callus differentiation: 1 mg L^−1^ BA, 0.5 mg L^−1^ NAA. For plantlets differentiation: 0.5 mg L^−1^ BA, 0.5 mg L^−1^ NAA	YL most effective for callus production. RL + BL (6:4) was the most effective for differentiating the highest number of plants per explant from embryogenic callus.	YL gave the highest values of callus fresh and dry weight, followed by RL + BL (60:40). This last light gave the highest values of plantlet height, fresh and dry weight.	[105]
*Vanilla planifolia* Andrews./Axillary buds axillary bud cuttings	25 µmol m^−2^ s^−1^ 16 h photoperiod	BL, RL, RL + BL (1:1), WL, Fl	9.55 µM BA	Fl, WL and RL + BL gave best results on shoot proliferation	Fl, WL and BL + RL determined higher shoot growth, plant height, leaves number, fresh weight, dry weight and chlorophyll content	[106]
*Gerbera jamesonii Bolus ex Hooker f.* cv Rosalin/In vitro propagated shoots	140 ± 10 μmol m^−2^ s^−1^	RL, BL, and their various mixtures. Fl was used as control	1 mg L^−1^ BAP and 0.1 mg L^−1^ NAA	Fl lamps, BL, WL and RL + BL (70:30) = the highest number of shoots/explant and 70% R + 30%.	The same treatments also yielded the highest values in terms of shoot length, plant fresh and dry weight.	[107]
*Cymbidium dayanum Rchb.f.* and *Cymbidium finlaysonianum Lindl.*/PLBs	50 μmol m^−2^ s^−1^ 16 h photoperiod	RL, BL, GL Fl.	(0, 0.1, 1 and 10 mg L^−1^), chondroitin sulfate	GL and BL + different concentrations of chondroitin sulfate promoted PLBs and shoots formation in the two species		[108]
*Bacopa monnieri* L. (Water hyssop)/Full, upper and lower, leaf cuttings.		WL, RL + BL (4:1, 3:1, 2:1,1:1)	0.25, 0.50 and 1.0 mg L^−1^ BA	WL was most effective in enhancing shoot regeneration.	Shoot length was increased by RL:BL (1:1) + 0.25 BA	[109]
*Vaccinium ashei* Reade cv Titan	50 µmol m^−2^ s^−1^ 16 h photoperiod	Fl, RL, RL + BL (80:20) (R8B2), RL + BL (50:50 (R5B5), BL.	1 mg L^−1^ zeatin riboside. Ventilated and non-ventilated vessels	No differences in shoot number between the different light treatments.	R8B2 and ventilated vessels were the most suitable for plant growth.	[110]
*Anthurium andreanum Lind*./Nodal segments	25 μmol m^−2^ s^−1^ 16 h light photoperiod	Fl, WL, RL, BL, BL + RL.	No growth regulators during the light treatments	BL + RL gave the highest number of adventitious shoots.	WL LEDs and BL LEDs, showed the greatest plantlet length and number of leaves. BL gave the greatest growth and chlorophyll content.	[111]
*Saccharum officinarum* L. variety RB867515)	(1) 72; (2) 60; (3) 53; (4) 77; (5) 46. 16 h photoperiod	BL:RL= (1) 70:30, (2) 50:50, (3) 30:70, (4) WL, (5) Fl	1.3 μM BAP.	BL:RL = 50:50 promoted proliferation	BL:RL = 50:50 promoted the highest stem length, fresh mass production, leaf number.	[112]
*Staphylea pinnata* L./in vitro regenerated shoots	35 μmol m^−2^ s^−1^ 16 h photoperiod	Fl, RL + BL (50:50:1), RL + BL + FR (49:49:2) RL + BL + WL (40:40:20)	5 µM BA, 0.5 µM NAA	Treatment with RB and RBFR resulted in increased multiplication rate as compared to Fl.	RB and RBFR increased leaf chlorophyll content and carotenoids. RBW light increased the number of newly developed leaves.	[113]
*Stevia rebaudiana* Bertoni/Nodal segments measuring 0.5–1 cm in length	40–50 μmol m^−2^ s^−1^. 16 h light photoperiod	Fl (Control), BL, RL, RL + BL (1:1), WL	1 mg L^−1^ BA.	RL = higher proliferation rate	Under BL + RL, maximum shoot elongation and leaf number	[114]
*Vanilla planifolia Andrews*/Nodal segments measuring 0.5–1 cm in length	40 μmol m^−2^ s^−1^ 16 h light photoperiod	Fl (control) BL, RL, RL + BL (1:1), WL	2.1 mg L^−1^ BA	No differences in shoot multiplication.	BL enhanced leaf number and area. RL + BL enhanced shoot lengtht and chlorophyll content Fl determined higher fresh and dry weight and carotenoids.	[115]
*Dendrobium sonia*,/Mature PLBs	μmol m^−2^ s^−1^: W 17.7 B 22.5 Y 24.6 R 15.6 16 h photoperiod	WL (control), BL, YL, and RL.	11.1 μM BAP and 11.42 μM IAA	YL induced early PLB formation, shoot differentiation and initiation, higher number of shoots per explant.	Under YL, higher leaf area and fresh weight, longer shoots under the other lights.	[116]
*Nicotiana tabacum* L. and *Artemisia annua*/In vitro-grown plantlets	35 µmoles cm^−1^ s^−1^	WL, RL + BL (1:1), RL + BL (3:1) RL + BL (1:3)	no growth regulators	In *Nicotiana* more shoots under 1:1 RL + BL In *Artemisia* under RL + BL (3:1)	In both species, RL + BL (3:1) determined taller shoots, and higher fresh weight.	[34]
*Saccharum officinarum* var. RB98710 (Sugarcane)/shoot segments	50 µmol m^−2^ s^−1^ for FL, 80 µmol m^−2^ s^−1^ for LED 16-h photoperiod	Fl, WL, RL + BL (82:18).	For callus induction in the dark two substrates: C1 = 9 μM 2,4-D and 1.1 μM BA; C2 = 13.6 μM 2,4-D + 2.2 μM BAP. For shoot regeneration: hormone free medium.	LED were ineffective on somatic embryo regeneration but successful on shoot multiplication from somatic embryo.	Root length, number of leaves, shoot fresh and dry biomass did not differ between treatments.	[117] six subcultures
*Gerbera jamesonii* Bolus ex. Hook f. cv. Dura/in vitro propagated shoots	40 μmol m^−2^ s^−1^ 16-h photoperiod	BL, RL + BL1 (50:50), RL + BL2 (70:30), RL + BL + WL (40:40:20), RL + BL + FR (49: 49:2), RL, Fl (Control)	5 μM BA (1,1 mg L^−1^) and 0.5 μM NAA (0.1 mg L^−1^)	RB1 and RB2 determined a higher shoot multiplication rate as compared to the control	RL = the greatest shoot elongation; BL = the highest leaf dry weight; RB2 = higher concentrations of total chlorophyll and carotenoids; RB1 = high leaf number.	[118]
*Lippia gracilis* Schauer./Apical and nodal segments	42 μmol m^−2^ s^−1^ 16 h photoperiod	WL, RL, BL, RL + BL (2.5:1 and 1:2.5)	no growth regulators	No influence of the light intensity nor of quality on shoot number both on nodal and apical segments.	RL and WL = best results on leaf and dry weights. B = higher photosynthetic pigment production in plantlets from apical explants, WL of those from nodal explants.	[119]
*Myrtus communis* L./Axillary shoots	35 µmol m^−2^ s^−1^ 16 h photoperiod	BL; RL:BL (70:30); RL; Fl = control.	0.5 μM L^−1^ NAA and different concentrations of BA: 1, 2.5 and 5 µM.	RL and 5 µM BA resulted in the highest multiplication rate.	At 5 µM BA, RL determined the higher dry weight; BL = a greater leaves number, BL and RL:BL increased the FW compared to Fl.	[120]
*Chrysanthemum × morifolium Ramat.*, *Ficus benjamina* L., *Gerbera jamesonii Bolus f.*, *Heuchera hybrida*, *and Lamprocapnos spectabilis* *(L.) Fukuhara*.	62–65 µM m^−2^ s^−1^ 16 h photoperiod	Fl (control), *NS1* lamps (BL + GL + RL + FRL- 21:38:35: 6) *G2 lamps* (BL + GL + RL+ FRL- 8:2:65:25), *AP673L* (BL + GL + RL + FRL- 12:19:61:8), *AP67* (BL + GL + RL + FRL-14:16: 53: 17)	No PGRs for *C. grandiflorum*; 4.0 mg L^−1^ BA and 30 mg L^−1^ adenine sulfate for *F. benjamina*; 3.0 mg L^−1^ K. for *G. jamesonii*; 0.1 mg L^−1^ BA and 0.1 mg L^−1^ IAA for *H. hybrida*; 0.25 mg L^−1^ BA and 0.25 mg L^−1^ IAA for *L. spectabilis*	Except for *F. benjamina*, RL and G2 lamp gave highest or similar propagation ratios as compared to Fl. NS1 lamps was also efficient for *G. jamesonii*, *H. hybrida* and *L. spectabilis*	The highest chlorophyll content was recorded under Fl and AP673L in all species, in NS1 in two species.	[35]
*Oryza sativa* L. cultivar Nipponbare.	50 μmol m^−2^ s^−1^. 12 h photoperiod	Fl, BL BL:RL = 3:1 BL:RL = 1:1; B:R = 1:3; RL;	For callus induction:2.0 mg L^−1^ 2,4-D. For callus differentiation: 1.0 mg L^−1^ 2,4-D. For shoot differentiation 0.5 mg L^−1^ K, 2 mg L^−1^ BA, 0.25 mg L^−1^ NAA	BL = decreased time for callus proliferation, differentiation and regeneration, and highest frequency of plantlet differentiation, and regeneration.	BL:RL = 1:1 highest seedling growth, chlorophyll, and carotenoid contents and photosynthetic rates.	[121]

Abbreviations: white (WL), blue (BL), red (RL), far-red (FRL), dark (D), fluorescent light (Fl), NAA (1-Naphthaleneacetic acid), BA (6-Benzylaminopurine), IAA (Indole 3- Acetic Acid), 2,4-D (2,4-dichlorophenoxyacetic acid), PLB-Protocorm-Like Bodies.

**Table 2 plants-11-00844-t002:** Summary of the use of LED lighting in in vitro propagation of woody species.

Studied Species/Explant Type	Light Intensity and Photoperiod	Light Spectra	Growth Regulators in Medium	Results on In Vitro Proliferation	Morphogenetic Response	Authors and Year
*Pseudotsuga menziesii* Mirb. Douglas fir embryo	0.01–0.71 W/cm^2^ 16 h photoperiod	8 different narrow bandwidth Fl having maxima each at one of the following wavelengths 371, 420, 467, 504, 550, 590, 660, and 740 nm.	Embryo from seeds; For callus induction: 800 pg L^−1^ IAA, 1 mg L^−1^ IBA, 1 mg L^−1^ BA, 1 mg L^−1^ AS-isopentyladenine After four weeks, 0.5 mM BA and 0.25 mM zeatin were added. No growth regulators for growing buds.	Callus and adventitious bud formation on the embryo-derived callus was maximum at (0.42 mW/cm^−2^) under RL (660 nm).		[122]
Woody ornamental plants. Organogenesis (axillary bud proliferation)		Fl (control), high pressure sodium lamps (HPS), BL and RL	Light pipe modified growth chambers	HPS increased shoot number as compared to FL. RL increased shoot number over control.		[123]
*Spirea nipponica* Maxim/Shoot explants from 8 to 10 week-old stock cultures	WL: low fluence 15.0–23.0; high fluence 47.0–62.0 µmol m^−2^ s^−1^; RL+FR: low fluence) 8.7–15.9 µmol m^−2^ s^−l^ 16 h photoperiod	WL, RL + Fr	BA 0.25, 0.4, or 0.5 mg L^−1^.	RL + FR = improved proliferation especially by 0.5 Ba addition. RL + FR followed by high fluence WL improved proliferation at lower BA levels.	RL + Fr favourably influenced shoot length and growth	[124] No LEDS
‘Mr.S 2/5’ clone of *Prunus domestica Ehrh.*/Cuttings;	WL = 38.0 BL = 9.1 RL = 19.6 FR = 7.2 µmol m^−2^ s^−1^	WL BL RL FR	Ba 0.6 mg L^−1^	In intact cuttings, WL gave the highest shoot proliferation In decapitated seedlings, all lights gave 100% bud outgrowth.	BL and WL = a higher number of nodes; RL = longer internodes. Shoots produced in RL were longer in decapitated seedlings.	[125] all experiments were repeated twice
*Cydonia oblonga* Mill/Leaves from the second to the fourth node of the apical portion of in vitro shoots	BL, WL and RL = 20 ± 1; FR = 1.2 R + B 10 + 10 B + Fr= 20 + 1.2 Fr + RL = 0.5 + 1.6 (µmol m^−2^ s^−1^)	D, BL, WL, FRL, RL, RL+Bl, BL+FRL RL+FRL After All light treatments, further 20 days of WL light exposure.	4.7 µM K and 0.5 µM NAA	Somatic embryogenesis was highest under RL treatment.		[126] No LEDS
*Prunus avium* L. cv ‘Hedelfinger’and one of its somatoclones/Leaves	~9 µmol 16 h photoperiod	WL, RL, BL, FR, D	2 mg dm^3^ TDZ+ 2,4-D or IAA	WL and BL = the highest node number. BL and FR = the highest shoot outgrowth from buds.	RL = highest shoot length under. WL and BL and WL= high chlorophyll.	[127] no LEDS
*Malus domestica* [Suckow] Borkh. genotype MM106/Shoot tips from in vitro cultures	~40 μmol m^−2^ s^−1^ 16 h photoperiod	WL, RL, BL, GL, YL, UV-AL, D	8.86 (2 mg L^−1^) µM BA, 0.53 (0.06 mg L^−1^) µM Ga_3_, 0.3 µM (0.1 mg L^−1^) IBA	GL and WL gave the higher total number of shoots at the end of the fourth culturing cycle.	Leader stem height was greater under D, RL and YL.	[128] No LEDs Four cycles
*Populus alba × P. berolinensis*/Stems from in vitro shoots	40 µmol m^−2^ s^−1^ 16 h photoperiod	GL, RL, BL and YL. Fl (control)	0.02 mg·L^−1^ NAA, and 0.1 mg·L^−1^ TDZ.	Fl and YL exhibited better effects on shoot regeneration		[129] no LEDs
Musa spp. cv.’Grande naine’ AAA)/Meristematic shoot tips	40 μmol m^−2^ s^−1^ 16 h photoperiod	WL, Fl	16.8 μM BAP, 3.8 μM IAA, 1 mg L^−1^ on a temporary immersion system (TIS)	WL under TIS enhanced shoot proliferation.		[130]
*Populus x euramericana* selected clones, ‘I-476’ and ‘Dorskamp’/Petioles (5-mm long) from in vitro plants	60 μm m^−2^ s^−1^ 16 h photoperiod	Fl, BL, RL, RL +BL (1:1 and 7:3), and RL + BL + GL (7:2:1)	0.44 µM BA	Highest shoot regeneration on RL + BL (1:1) for ‘I-476’, on BL +RL (7:3) for *‘Dorskamp’* as compared to Fl.	High RL (100% or 7:3) = higher shoot length and leaf area BL or RL +BL (7:3) = higher stem diameter	[131]
*Malus domestica* [Suckow] Borkh rootstock cvs. Budagovsky 9 (B.9), Geneva 30 (G.30), and Geneva 41 (G.41). I exp = single-node segments	BL = 5.7 RL = 6.6 WL = 25 μmol·m^−2^·s^−1^	WL, RL, BL for both experiments	1.0 mg·L^−1^ BA, 0.1 mg·L^−1^ IBA, and 0.5 mg·L^−1^ GA_3_. II exp: cv. G.30 with and without gibberellic acid (GA3).	RL increased the number of shoots in B.9 and G.30 as compared to WL.	RL increased the length, and the number of elongated shoots of B.9 and G.30. GA_3_ promoted shoot growth of G.30 under RL and BL.	[132] No LEDS
*Phoenix dactylifera* L. cv. ‘Alshakr’ (Date palm)/shoot buds	20–25 μmol m^−2^ s^−1^ 14 h photoperiod	FL (control), RL +BL (18:2) (CRB-LED)	1 mg L^−1^ (NAA), 0.5 mg L^−1^ (BA) and 0.5 mg L^−1^ kinetin (K)	CRB enhanced the percentage of buds producing shoots and average shoots formation compared to FL	CRB-LED enhanced total soluble carbohydrates, starch, free amino acids, and peroxidase activity	[133]
*Camellia oleifera* C. Abel/Axillary buds	50 m^−2^ s^−1^ 16 h photoperiod	RL, BL, RL + BL, (4:1) RL + BL (1:4), WL was used as control	3.0 mg L^−1^ BA + 0.02 mg L^−1^ IBA	RL + BL (4:1) = the highest proliferation coefficient.	RL + BL (4:1) = good chlorophyll content, the thickest leaves, high stomatal density.	[134]

Abbreviations: white (WL), blue (BL), red (RL), far-red (FRL), dark (D), fluorescent light (Fl), NAA (1-Naphthaleneacetic acid), BA (6-Benzylaminopurine), IAA (Indole 3- Acetic Acid), 2,4-D (2,4-dichlorophenoxyacetic acid).

**Table 3 plants-11-00844-t003:** Effects of different light intensities on shoot proliferation in increasing light-intensity order.

Species	Tested Intensities	Best Yielding Intensity (μmol m^−2^ s^−1^)	Main Parameters Affected and Notes	Authors
*Disanthus cercidifolius*, *Rhododendron* spp., and *Crataegus oxyacantha*	11, 25, 55, 106 and 161 µmol m^−2^ s^−1^	11–27	Better growth and leaf chlorophyll content	[75]
*Acer saccharum* *Marshall*	4, 16 and 40 µmol m^−2^ s^−1^	4 and 16	Low intensity overcomes recalcitrance.	[214]
*Achillea millefolium* L.	13; 27; 35; 47 and 69 µmol m^−2^ s^−1^	27 µmol m^−2^ s^−1^	Higher dry mass of shoots and roots, shoot length	[172]
*Withania somnifera* (L.)	15, 30, 60, and 90 µmol m^−2^ s^−1^	30 μmol m^−2^ s^−1^	Greater growth and development.	[182]
*Chrysanthemum morifolium* Ramat. ‘Ellen’	25, 40, 55, 70, 55 µmol m^−2^ s^−1^	40 µmol m^−2^ s^−1^	Better plantlet growth	[174]
*Vaccinium corymbosum*)	55 to 240 µmol m^−2^ s^−1^ for 7 to 60 days		Higher irradiances (≥55 = 210 µmol m^−2^ s^−1^) improved proliferation only with short time applications (7 days).	[215]
*Spathiphyllum cannifolium* Culture Pack”, on rockwool system, with CO_2_ enrichment	45, 60, 75 μmol m^−2^ s^−1^ 80% RL + 20% BL LED	60 μmol m^−2^ s^−1^	Best growth	[216]
*Fragaria* × *ananassa* Duchesne	45, 60, 75 µmol m^−2^ s^−1^	60 μmol m^−2^ s^−1^	Better shoot growth	[149]
*Plectranthus amboinicus* (Lour.) Sprengof	26, 51, 69, 94 and 130 μmol m^−2^ s^−1^	69 µmol m^−2^ s^−1^ and to a lesser extend 94	Higher shoot number, leaf area, total dry weight and carvacrol content	[48]
*Phaius tankervilliae* (Banks ex L’Herit) and *Vanda coerulea* Giff	28, 37, 56, 74 and 93 μmol m^−2^ s^−1^	74 µmol m^−2^ s^−1^	Better plantlet growth	[217]
*Pyrus* spp. *rootstock* BP10030	from 10 to 80 μmol m^−2^ s^−1^ 16 and 24 h photoperiod	from 10 to 80 μmol m^−2^ s^−1^ 16 h photoperiod = greatest shoot number	10 μmol m^−2^ s^−1^ better for initial explant growth. Increasing irradiance to max higher growth 24 h = the highest shoot fresh and dry weight.	[218]
*Lippia gracilis Schauer*	26, 51, 69, 94, or 130 μmol m^−2^ s^−1^	94 µmol m^−2^ s^−1^	higher number of segments, leaf, shoot, root, and total weight plantlet^−1^	[119]
*Momordica grosvenorii Swingle*	25, 50, 100, or 200 μmol·m^−2^·s^−1^, and an increased CO_2_ concentration	increasing intensities up to 100 µmol m^−2^ s^−1^	Better plantlet growth	[219]
*Actinidia deliciosa (A. Chev.) C.F. Liang & A.R.*	30 to 250 μmol m^−2^ s^−1^ and an increased CO_2_ concentration	120 μmol m^−2^ s^−1^	better plantlet growth and proliferation	[220]
*Rosa hybrida*	0, 4, 17, 66, and 148 μE m^−2^ s^−1^	17 μE m^−2^ s^−1^ 148 μE m^−2^ s^−1^	At the highest intensity best proliferation. At 17 μE m^−2^ s^−1^ lower propagation but better leaves	[221]

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
