# Peer review of "Light and Plant Growth Regulators on In Vitro Proliferation"

_plants, 2022, doi:10.3390/plants11070844_

Round 1

Reviewer 1 Report

Plants 2022 doi 10.3380/xxxxx. Cavarallo et al.

A comprehensive review.

Perhaps it is unavoidable, granted the subject matter, that few clear conclusions can be drawn.

It would be helpful if all plants could be referred by their Linnean Binomials (in italics) with, where appropriate cultivar names. Furthermore, when oxygenic photosynthetic organisms other than flowering plants are mentioned, e.g. Chlorella, their taxonomonic status should be mentioned, In the case Chlorella this is Chlorophyta: Trenouxiophyceae.

Lines 28, 36. ‘that’, not ‘to’. Avoid teleology.

Line 47. ‘mitochondria’.

Lines 68-69. Clarify ‘subject to scarce modification, thus blowing up…’.

Line 74. Clarify ‘occur for substein the adaptation.

Line 78. How rapidly can evolution occur in terms of number of plant populations?

Line 92. ‘in this context’, not ‘under this contest’.

Lines 93-96. This list has 4 external factors and 1 physiological outcomes. Clarify.

Line 100. ‘and for a long time’ is superfluous.

Line 102. State what wavelength outside 400-700 nm the light sources provide.

Lines 113-114. Explain maximum PAR efficiency of 100%. This implies perfect matching of the photon output of the lamps and the absorption spectrum of the plants on a photon basis.

Line 121. Are there changes in the spectrum and energy output of LEDs through their life?

Line 123. Provided electricity comes from renewables.

Lines 164-165. These three categories cover all flowering plants on land.

Line 178. Delete ‘the’.

Line 193. Chlorophyll.

Line 209. ‘auxins’: dos this imply substances in adition to indoleacetic acid?

Lines 238-239. ‘and on’ repeated.

Lines 270-271. Chlorophyll.

Lines 336. SEM stands for Scanning Electron Microscopy.

Line 365. Delete ‘light’.

Line 416. ‘In contrast’, not ‘On the opposite’.

Line 439. ‘weight’.

Line 553. Rephrase to avoid the implication that chlorophyll a absorbs only at 460 nm and chlorophyll b absorbs only at 660 nm.

Line 642. ‘cytokinin’, not ‘cytokine’.

Line 663. ‘it is a’.

Line 668. ‘absorption’.

Table 3, last entry. What does ’17 per square metre per second’ .                        

Line 999. ‘strigolactones’.

Line 1022. ‘It has been known for several years’

Line 1097. Clarify’.

Author Response

We wish to thank you for your comments which helped us to improve our paper.

You will find the reviewer comments in italics and ours in normal font.

A comprehensive review.

Perhaps it is unavoidable, granted the subject matter, that few clear conclusions can be drawn.

Yes, it is true, even because we try to highlight the lack of a methodology which may improve the uniformity and the reliability of the results. However, in the discussion and conclusions we tried to give some indications concerning the light intensity, photoperiod and spectrum. In relation to this last, we stress the opportunity of using a combined light (white) properly integrated or modified.

It would be helpful if all plants could be referred by their Linnean Binomials (in italics) with, where appropriate cultivar names. Furthermore, when oxygenic photosynthetic organisms other than flowering plants are mentioned, e.g. Chlorella, their taxonomonic status should be mentioned, In the case Chlorella this is Chlorophyta: Trenouxiophyceae

Done

Lines 28, 36. ‘that’, not ‘to’. Avoid teleology.

Done

Line 47. ‘mitochondria’.

We changed with mitochondrion

 Lines 68-69. Clarify ‘subject to scarce modification, thus blowing up…

Done

Line 74. Clarify ‘occur for substein the adaptation.

Done

Line 78. How rapidly can evolution occur in terms of number of plant populations?

We are really sorry, but we haven’t catch up the meaning of the questions. Usually, plantlets cultured in vitro is not a population of individual genotype, but it is a set of clonal plantlets.

Line 92. ‘in this context’, not ‘under this contest’.

Done

Lines 93-96. This list has 4 external factors and 1 physiological outcomes. Clarify.

We added: ‘i.e. the species-specific physiological adaptation to in vitro conditions previously described’

 Line 100. ‘and for a long time’ is superfluous.

Deleted

 Line 102. State what wavelength outside 400-700 nm the light sources provide.

Since most authors report that fluorescent lamps provide wavelengths in the 400- 700 nm range, Dutta Gupta and Jatothu (2013, n.41 in our bibliography) precise that they have wavelengths ranging from 350 to 750.

We have also inserted the following information in the text: ‘The ILs have high amounts of photons in the infrared and red ranges, gradually dropping toward blue. Due to the presence of phosphor coating, white FLs also have a continuous visible spectrum with peaks near 400–450 nm (violet-blue), 540–560 nm (green-yellow), and 620–630 nm (orange-red)’.

 This phrase is now shifted from 114 to 117 line.

Lines 113-114. Explain maximum PAR efficiency of 100%. This implies perfect matching of the photon output of the lamps and the absorption spectrum of the plants on a photon basis.

We have uploaded this information from the reference [45].

Line 121. Are there changes in the spectrum and energy output of LEDs through their life?

We added in the text the lifespan of LED lamps.

Line 123. Provided electricity comes from renewables.

 Lines 164-165. These three categories cover all flowering plants on land.

Corrected: “ in flowering plant species”

Line 178. Delete ‘the’.

Done

Line 193. Chlorophyll

Done

Line 209. ‘auxins’: does this imply substances in addition to indoleacetic acid?

Corrected with auxin

Lines 238-239. ‘and on’ repeated.

Done

Lines 270-271. Chlorophyll.

Done

Lines 336. SEM stands for Scanning Electron Microscopy.

Corrected

Line 365. Delete ‘light’.

Done

Line 416. ‘In contrast’, not ‘On the opposite’.

Done

Line 439. ‘weight’.

Done

Line 553. Rephrase to avoid the implication that chlorophyll a absorbs only at 460 nm and chlorophyll b absorbs only at 660 nm.

We have not found the line related with the arguments discussed by the referee.

Line 642. ‘cytokinin’, not ‘cytokine’.

Done

Line 663. ‘it is a’.

Done

Line 668. ‘absorption’.

Done

Table 3, last entry. What does ’17 per square metre per second’ .                       

Corrected with 17 μE m-2 s-1 as reported in the article

Line 999. ‘strigolactones’.

Done

Line 1022. ‘It has been known for several years’

Done

Line 1097. Clarify’.

Corrected

Best regards

The authors

Reviewer 2 Report

The article "Light and plant growth regulators on in vitro proliferation" is a nice theme for review. However, there is a serious problem with the structure of the review. Authors should put a proper subheading and put the relevant information under that heading and subheading.  Apart from this,  it lacks flow.  The Authors should revise critically. For example in the introduction, at lines 59 -60, the Authors talk about PHOT1 and 2 receptors and directly jump to micropropagation at line 60. Further review should be crisp and precise.

In Table 1, why authors only summarised LED light, not cool light ? Table 1 is just a collection of various literature not meaningful. The  current form of review is written in hurry and therefore it must be rejected. Authors are encouraged to submit it fresh after extensive revision.

Author Response

The authors wish to thank you for the comments to our paper.

You will find thereafter your suggestions in Italics, ours in normal font.

The article "Light and plant growth regulators on in vitro proliferation" is a nice theme for review. However, there is a serious problem with the structure of the review. Authors should put a proper subheading and put the relevant information under that heading and subheading.  Apart from this, it lacks flow.  The Authors should revise critically. For example in the introduction, at lines 59 -60, the Authors talk about PHOT1 and 2 receptors and directly jump to micropropagation at line 60. Further review should be crisp and precise.

We have re-organized the heading and subheading of manuscript following the suggestion of the journal.

We have also modified some periods to make the reading clearer.

In the paragraph where PHOT1 and 2 are mentioned together with the other photoreceptor; in fact, the function of this small paragraph should be intended as a small list of the main plant photoreceptors. We understand the referee's misgivings but feel that the break is necessary to direct the reader immediately to the context of the biological system that is the subject of the review.

 In Table 1, why authors only summarised LED light, not cool light? Table 1 is just a collection of various literature not meaningful.

Now table 1 moved to the supplementary materials. However, with this table we aimed at summarizing the most of the protocols and results of the experiments carried out in vitro on light effects on in vitro proliferation.  In some of the references we indicated that LEDS are not used (see for example references 67, 68, 69, 70) in this cases white light is represented by cool fluorescent light, in some others cool fluorescent light (Fl,  see 72, 74, 76 and so on) is used as the control light.

The  current form of review is written in hurry and therefore it must be rejected. Authors are encouraged to submit it fresh after extensive revision.

We revised all the manuscript.

Reviewer 3 Report

A comprehensive review on light quality (but not much on quantity) and in plant growth, I enjoy reading it, thank you.

A few minor things could be improved:

1, The numeration and organisaiton of subtitles is not friendly. And why the white light is in the middle of some monochromatic lights?

2, “3. Effects of light intensity” (the font is not in bold), authors are aware of the fact that artificial lights are much lower in intensity comparing to natural light, it would be better to clarify how big the difference is. In Table 3, the highest light intensity cited is 200umol m-2 s-1 for Momordica grosvenori, how is the normal light intensity in the native habitat of this plant? As far as I know, a measurement of thousands umol m-2 s-1 of light intensity can be easily achieved at outdoor in a sunny day even in London.  Photoinhibition is part of normality for plants growing outdoor and yet is ingored in most in vitro proliferation. 

3. line 1046, is this meant to be a new subtitle?

Author Response

We wish to thank you for your  appreciation and comments which helped us to improve our paper.

You will find the reviewer comments in italics and ours in normal font.

A comprehensive review on light quality (but not much on quantity) and in plant growth, I enjoy reading it, thank you.

A few minor things could be improved:

1, The numeration and organization of subtitles is not friendly. And why the white light is in the middle of some monochromatic lights?

The white light quality is mentioned only when there is the comparation to highlight the different effect of this combined light with the monochromatic lights.

2, “3. Effects of light intensity” (the font is not in bold), authors are aware of the fact that artificial lights are much lower in intensity comparing to natural light, it would be better to clarify how big the difference is. In Table 3, the highest light intensity cited is 200umol m-2 s-1 for Momordica grosvenori, how is the normal light intensity in the native habitat of this plant? As far as I know, a measurement of thousands umol m-2 s-1 of light intensity can be easily achieved at outdoor in a sunny day even in London.  Photoinhibition is part of normality for plants growing outdoor and yet is ignored in most in vitro proliferation.

Since the beginning of the plant tissue culture appeared clear that light intensity, independently from the type of light source, was inhibitor of plantlets growth. In fact, in all tested plant species light intensity higher of the specific threshold induced lignification of stem plantlets and development of very large leaf as could be comparable with in vivo growing plants. Moreover, it has been demonstrated that too high light intensity (>200 μmol m-2 s-1 in the most resistant genotypes) induces oxidative stress damages.

3. line 1046, is this meant to be a new subtitle?

Yes. Corrected

Best regards 

The authors

Reviewer 4 Report

I completed the review of the article: Light and plant growth regulators on in vitro proliferation.
It is a topical review on the influence of light on plant micropropagation in vitro. The strong points are the systematized information and the discussion depending on the type of light so this review will be useful both for researchers who want to improve the protocol of propagation and for the industry because, if properly used, LED light can bring benefits by reducing energy costs, improve the results, and a better quality. I have some specific comments that only relate to spelling considering that there is a lot of information.
L26-28 surrounding environment is used twice, a synonym should be used.
L 44, responses not responces
L 53 I think the expression should be improved
L140-142 I think the expression should be improved
L175 I think that the subtitles everywhere in the review ex ''Adventitious shoot number'' should have a subchapter number of ex 2.1.1

Author Response

We wish to thank you for your appreciation and comments which helped us to improve our paper.

You will find the reviewer comments in italics and ours in normal font.

I completed the review of the article: Light and plant growth regulators on in vitro proliferation.

It is a topical review on the influence of light on plant micropropagation in vitro. The strong points are the systematized information and the discussion depending on the type of light so this review will be useful both for researchers who want to improve the protocol of propagation and for the industry because, if properly used, LED light can bring benefits by reducing energy costs, improve the results, and a better quality. I have some specific comments that only relate to spelling considering that there is a lot of information.

L26-28 surrounding environment is used twice, a synonym should be used.

Surrounding was cancelled

L 44, responses not responses

Done

L 53 I think the expression should be improved

Corrected

L140-142 I think the expression should be improved

Improved

L175 I think that the subtitles everywhere in the review ex ''Adventitious shoot number'' should have a subchapter number of ex 2.1.1

Done

Best regards 

The authors